



# Landslide hazard probability and risk assessment at the community level: a case of western Hubei, China

**Sheng Fu[1], Lixia Chen[1], Tsehaie Woldai[2], Kunlong Yin[3], Lei Gui[3], Deying Li[3], Juan Du[4], Chao Zhou[5], Yong Xu[6], and Zhipeng Lian[6]**

[1]Institute of Geophysics and Geomatics, China University of Geosciences, Wuhan 430074, China
[2]School of Geosciences, University of Witwatersrand, Johannesburg 2050, South Africa
[3]Engineering Faculty, China University of Geosciences, Wuhan 430074, China
[4]Three Gorges Research Center for Geohazards, China University of Geosciences, Wuhan 430074, China
[5]School of Geography and Information Engineering, China University of Geosciences, Wuhan 430078, China
[6]Wuhan Centre of China Geological Survey, Wuhan 430205, China

**Correspondence:** Lixia Chen (lixiachen@cug.edu.cn)

**Abstract.** Small communities living in mountainous terrain in Hubei province are often affected by landslides. Previous studies by the China Geological Survey focused on the 1 : 100 000 scale. Therefore, a more detailed assessment, especially at the community level, is urgently required by local governments for risk management. In this study, we conducted a more detailed semiquantitative landslide and risk assessment at the community level using a scale of 1 : 10 000. We applied the probabilistic method to assess landslide spatial, temporal, and size probabilities, while the hazard and risk assessment were considered for four return periods (5, 10, 20, and 50 years) and two size scenarios (landslide volume). The spatial probability from susceptibility mapping with an accuracy of 84 % indicates that the major controlling factors are Quaternary deposits and weathered eluvium from Ordovician limestones. This study revealed that most building areas in hazard maps are at the foot of major slopes with very high hazard probabilities, and therefore we computed the potential loss of life and property for each slope. The results reveal that 1530 people and USD 18 million worth of property were at risk of landslides within a 50-year return period and a landslide volume of 50 000 m$^3$. The longer the return period is, the higher the hazard probability is. Compared with the classic inverse gamma and power law distribution of landslide magnitude and frequency, the function by the ordinary least squares method is more suitable for landslide size probability analysis of the study area. According to

these methods, the proposed procedure of landslide risk assessment proves more useful than the existing data from the 1 : 100 000 scale in western Hubei, China.

## 1 Introduction

Risk analysis and assessment in China is an effective means of reducing casualties and economic losses induced by landslides. Although theory and techniques applied worldwide are available (Van Westen et al., 2005; Lee et al., 2007; Neuhäuser and Terhorst, 2007; Erener et al., 2016; Huang et al., 2017; Jiménez-Perálvarez et al., 2017; Van Westen and Greiving, 2017), these are not yet well utilized in western Hubei, China, where the current research was undertaken. To date, very little scientific work is documented at the community level for the study area. Mountainous communities in the area are exposed to landslides because of high rainfall and urbanization. Annually, road construction and anthropogenic modifications at the community level (e.g., excavations in search of building materials), the degree of urbanization, and subsequent population growth have promoted the frequency of landslides, with immense risk to the communities, causing death and unaccountable property loss. According to the Chinese geological disaster notification report (2017), for example, around 850 disasters occurred in 2017 in Hubei province, China, causing 23 deaths and about CNY 254 million (equal

to USD $\sim$ 36.29 million by November 2019 conversion) of economic losses. Quantifying landslide risk and developing a reduction strategy remains a challenging issue.

Currently, guidelines for landslide risk zoning and land use planning with the framework, definitions, and recommendations are available for clearly defined scales (Fell et al., 2008a). Also, research work highlighting landslide risks at a community level has recently been conducted, and some of the results reached are available to the public (Erener and Düzgün, 2012; Abdulwahid and Pradhan, 2016; H. X. Chen et al., 2016; Liu et al., 2016; McAdoo et al., 2018; Paliaga et al., 2019). Paliaga et al. (2019), for example, used a spatial multicriteria analysis technique to propose geohydrological risk mitigation measures in a small but densely populated catchment, with descriptive parameters involving the extent of urban development and elements at risk. In Spain, a quantitative assessment of landslide risk for the road network of the Basque Country was used for calculating hazard probability and expected consequences (Mavrouli et al., 2019).

Our objective in this study is to conduct a community level landslide hazard probability and risk assessment of an area with limited landslide data and damage records. Despite these limitations, in this study, we attempted to quantify the landslide risk for the Yuyangguan community, in Hubei province, China. The probabilistic method was used to assess landslide spatial, temporal, and size probabilities. Landslide hazard and risk assessment are considered for four return periods and two magnitude scenarios. From these, we propose risk reduction strategies that stakeholders can use for risk management and control.

## 2   Study area and data

The study area, Yuyangguan community (30°07′–30°12′ N, 111°01′–111°07′ E), is in Wufeng county (29°56′–30°25′ N, 110°14′–111°20′ E), western Hubei province, China (Fig. 1). It covers an area of about 34 km$^2$ with 15 villages, including Sanfangping, Dafangping, and Caojiaping. This area has been inhabited since 750 years ago, but intense urbanization development involving the construction of national-class roads began only in 2012. The area was selected for this study because of frequent landslides responsible for tremendous damage in recent years. The residential area is surrounded by steep slopes, with elevations ranging from 180 to 680 m a.s.l. The climate is typical monsoonal with annual average precipitation of about 1500 mm.

The main lithological units outcropping in the study area comprise Ordovician ($O_2g$) limestones and shales, Silurian ($S_2s$) sandstones and shales, Devonian sandstones with coal layer, Permian sandstones, and recent deposits. The $S_2s$ sandstones dominate the rock types along the community slopes, and intense weathering explains the low mass strength of the bedrock. Consequently, weathered $S_2s$ and

$O_2g$ rocks are the primary sources of landslides, with the records and bedrocks presented in Table 2.

The Xiannvshan fault (Fault I in Fig. 1) and Yuyangguan fault (Fault II in Fig. 1) of the Changleping tectonic belt constrain the bulk architecture of the area. The Xiannvshan fault, striking approximately 340° N–345° W and dipping 60–70° NE, is a transpressional fault terminating at the center of the community. The fault is a seismically active belt, exemplified by the Panjiawan earthquake ($M_s = 4.9$) of 1961 and the Zigui earthquake ($M_s = 3.3$) of 1972. Conversely, the Yuyangguan fault, striking E–W and dipping 60–70° S, is a transtensional fault. Rocks associated with the Yuyangguan fault comprise a 20–50 m wide area of cataclasite and brecciated mylonite, with several secondary faults merging into it, and, together, go through the center of the Yuyangguan community. According to the China Earthquake Administration, the studied area is in a weak-seismic-activity region with a basic earthquake intensity of VI, and the 50-year 10 % probability exceedance of the peak ground acceleration is 0.05 g. No historical record is available for earthquake-induced landslides in the area.

Rainfall and anthropogenic activities contribute significantly to triggering mass movements, mainly landslides. In the rainy season of 2013, a slope along the main road collapsed, breaking and causing lengthy traffic jams and transportation problems (Fig. 2). A landslide occurred in a new residential quarter of the community on 3 January 2013, due to an unstable foundation pit after a 1 d excavation (Fig. 3). These two landslides are examples, with more landslide records prepared from aerial photograph interpretations and validated by field investigations.

Table 2 shows all historical landslide data from 1976 to 2013 in areas dominated by Quaternary deposits. Besides the landslide inventory database, other datasets collected for landslide risk mapping include the following.

- A 10 m × 10 m resolution digital elevation model (DEM) generated from a topographic map obtained using an unmanned aerial vehicle (UAV) was used. The DEM allowed the extraction of the slope, elevation, aspect, and curvature data using the surface analysis tool in ArcGIS (a geographic information system for working with maps and geographic information, http://www.esri.com/software/arcgis/arcgisonline, last access: September 2017; Fig. 4a–d).

- A Geological map at a 1 : 50 000 scale (Fig. 1) was used to extract datasets, including lithology, faults, and slope structure. The slope structure map (Fig. 4e) was generated using the standard and stratigraphic altitude advocated by Cruden (1991). The land use map provided the distribution of rivers and roads (Fig. 4h and i).

- Landslide inventory databases of the case community (named Yuyangguan) and a neighboring community (named Chengguan) were utilized in this study to an-

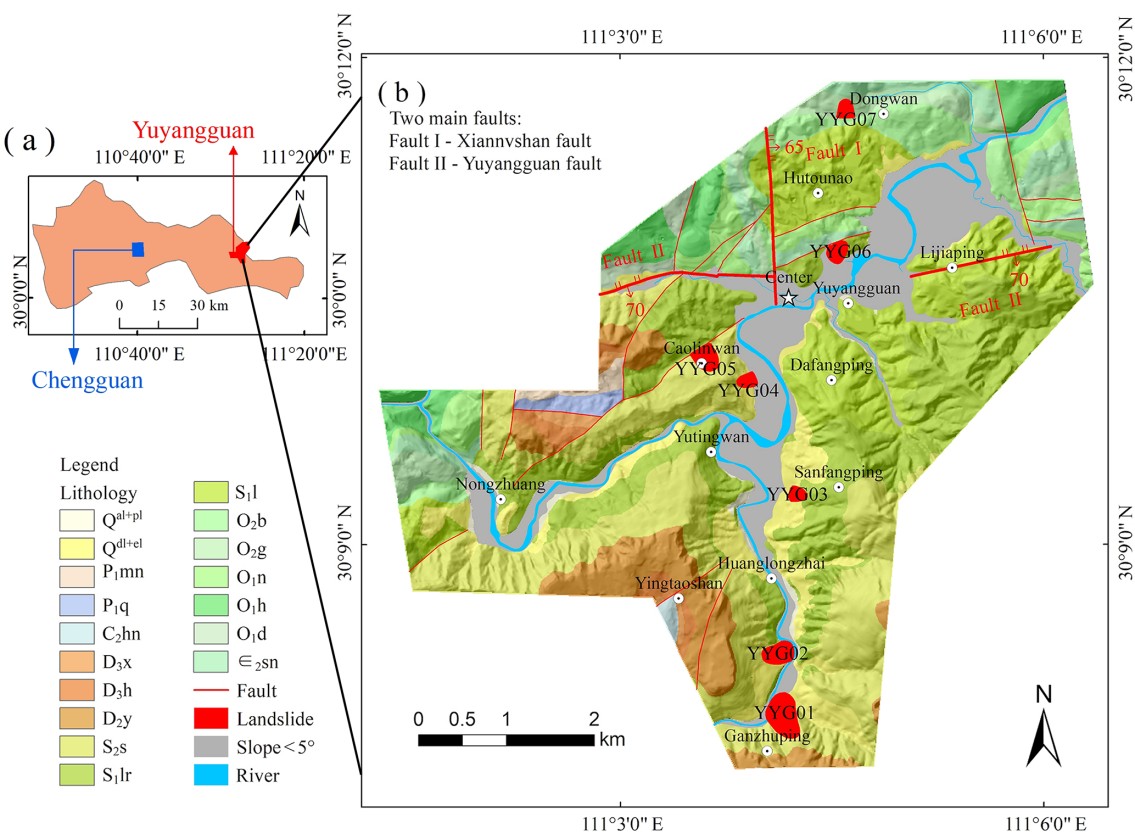

**Figure 1. (a)** Insert representing Wufeng county hosting the Yuyangguan and neighboring Chengguan community; **(b)** the distribution of lithological units and recorded landslides in the case community. (The lithology codes in the legend are described in Table 1.)

**Table 1.** Characteristics of lithology distributed among Yuyangguan.

| Lithology code | Characteristics of lithology |
|---|---|
| $Q^{a_l+p_l}$ | Gravel, pebble, and drift stone, with a small amount of sand. |
| $Q^{d_l+e_l}$ | Clay intercalated with gravel, mainly distributing in the gentle slope area of the bank slope. |
| $P_1mn$ | The top marl, the middle thin-layered manganese-bearing siliceous limestone. |
| $P_1q$ | Upper Carboniferous tumorous limestone, lower chernite nodules, and chernite strip limestone. |
| $C_2hn$ | Upper thick-layered limestone, dolomitic limestone, lower dolomite, sometimes conglomerate. |
| $D_3x$ | Upper sandstone and shale interbedded, middle thick-layered marl, lower sandy shale with oolitic hematite. |
| $D_3h$ | Thin, medium-thick silty shale, fine-grained quartz sandstone, bottom shale. |
| $D_2y$ | Thick quartzite and quartz sandstone with a small amount of carbonaceous shale and mudstone shale. |
| $S_2s$ | Thick- to thin-layered quartz sandstone, siltstone and silty shale, and mudstone shale at the lower part. |
| $S_1lr$ | Shale with siltstone and thin marl. |
| $S_1l$ | Muddy shale, sandy shale with siltstone, silty shale, and carbonaceous shale. |
| $O_2b$ | Medium-thick-layered bioclastic turtle limestone. |
| $O_2g$ | Microcrystalline limestone with a medium thickness. |
| $O_1n$ | Medium- and thick-layered limestone, dolomite and shale with limestone at the bottom. |
| $O_1h$ | Thick and massive coarse-grained bioclastic limestone and limestone. |
| $O_1d$ | Thick to thin layers of tumorous limestone interbedded with shale. |
| $\epsilon_2sn$ | Massive and thick-layered dolomite with dolomitic limestone. |

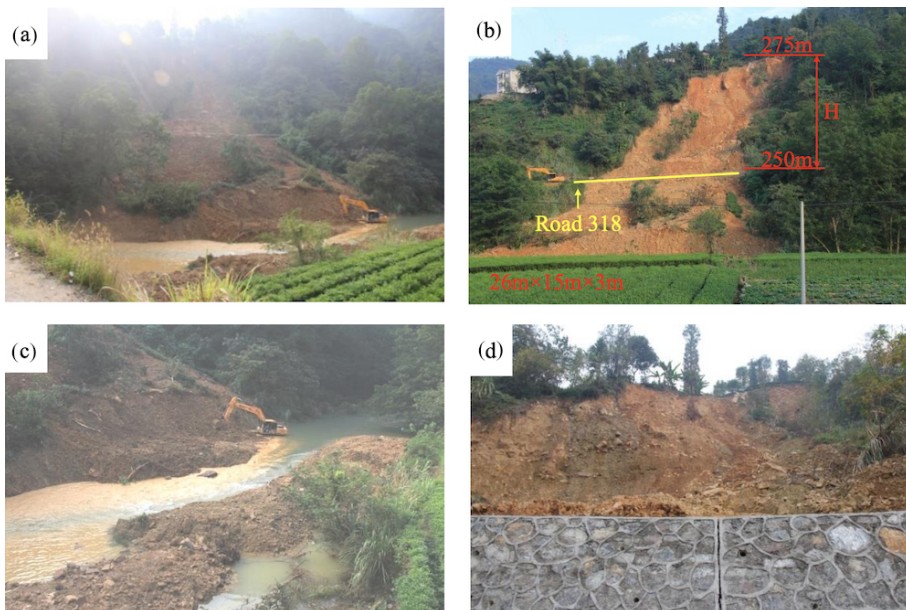

**Figure 2.** Landslide YYG01 occurred in the rainy season of 2013, causing damage on the national road (G318) in Yuyangguan. (Travel distance = 57 m.) **(a)** An overview of the landslide YYG01, **(b)** left side of the loose landslide body, **(c)** river blockage caused by the landslide YYG01, and **(d)** the retaining wall.

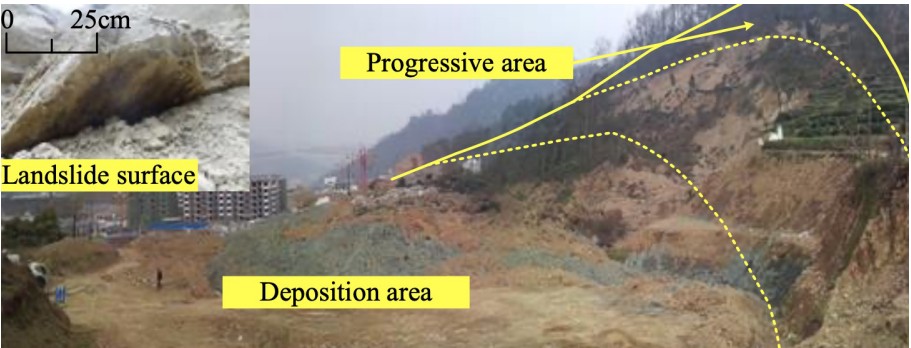

**Figure 3.** Landslide YYG03 caused by slope incision in a residential quarter of Yuyangguan.

alyze the size probability in hazard assessment (see Table 2). We used the hazard database of the neighboring community for the following reasons: (1) the landslides in the case community are limited for probability analysis; and (2) the two communities are similar in geomorphology, geology, climate, and landslide types. The location of the Chengguan community is displayed in Fig. 1a, and administratively the two communities are in Wufeng county. Both communities belong to the same structural belt named the Changleping anticline fold that extends in a nearly E–W direction and comprises the Silurian and Ordovician rocks. Previous landslides in these communities involve shallow and soil slope movement of weathered bedrock. The factors triggering the landslides are rainfall and slope cutting, associated with urbanization development.

– A building footprint map (Fig. 5) was interpreted and checked in the field by the authors, with most buildings on or at the toe of the first slope zones, with an elevation of up to 350 m a.s.l. The entire built area is 757 000 m$^2$, with data on the economic value of buildings obtained from the Department of Natural Resources of Hubei Province (see Table 3).

– Census data were obtained by integrating the information derived from the China population data (2010) (http://www.stats.gov.cn/tjsj/pcsj/rkpc/6rp/indexch. htm, last access: September 2016) and sampling survey (Fig. 6), amounting to a population of 45 914 in the area.

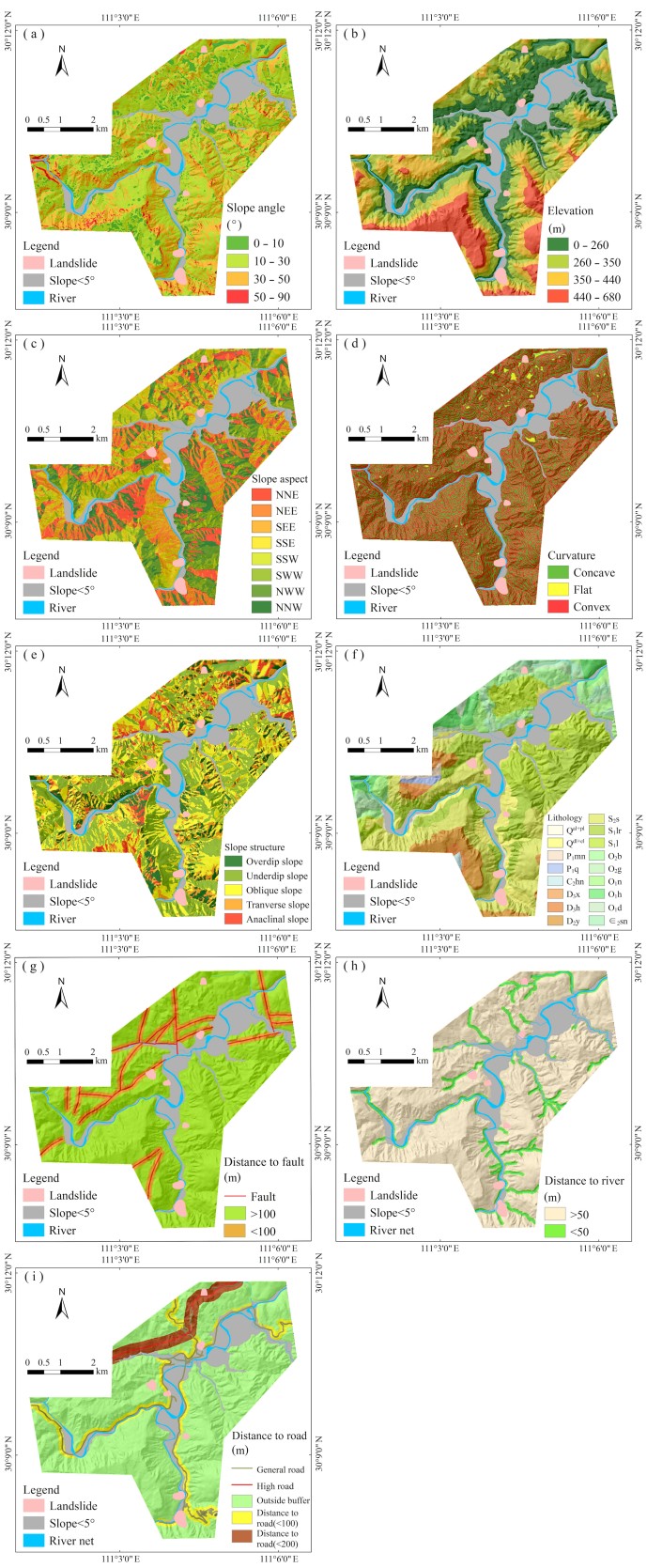

**Figure 4.** Thematic maps for landslide susceptibility mapping: **(a)** slope, **(b)** elevation, **(c)** slope aspect, **(d)** curvature, **(e)** slope structure, **(f)** lithology, **(g)** distance to fault, **(h)** distance to river, and **(i)** distance to road.

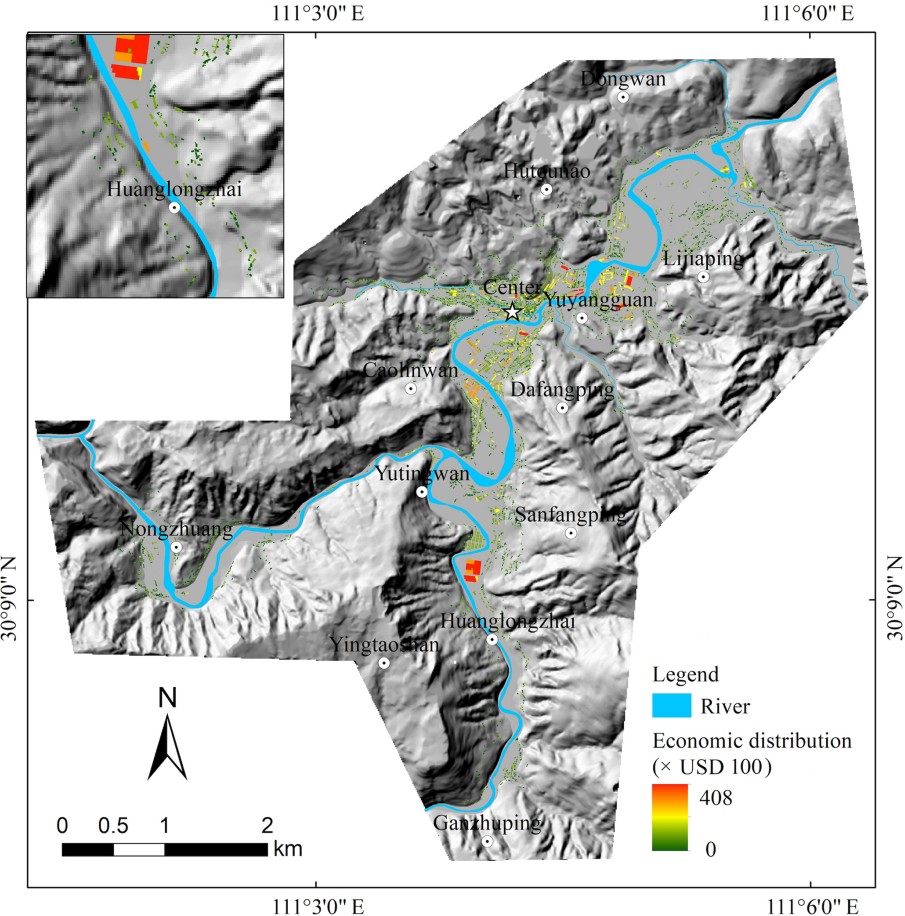

**Figure 5.** Building footprint map with economic values for the Yuyangguan community.

## 3 Methodology

This section summarizes the methods employed for semi-quantitative risk analysis for landslides at a community level. Initially, the slope-unit-based hazard probability was calculated by integrating spatial, temporal, and size probabilities. We generated eight hazard probability maps involving four return periods (5, 10, 20, and 50 years) and two size scenarios. For each map, we determined the potential landslide-influencing area semiquantitatively by calculating the traveling distance of the slope unit. Subsequently, the element-at-risk map was interpreted from the image data, population census data, and sampling survey. The vulnerability map was created by assimilating assets and landslide-influencing areas. Values in the vulnerability map were semiquantitatively determined for buildings and people in the buildings. The combination of hazard maps, vulnerability maps, and element-at-risk maps produced the risk value for each slope unit, contributing to the final risk maps of the population and buildings in the study area. The flowchart of the methodology is depicted in Fig. 7.

### 3.1 Landslide hazard from spatial, temporal, and size probability analysis

Hazard assessment is an essential step in landslide risk assessment. For the community level, this is achievable through the deterministic model (Gokceoglu and Aksoy, 1996; Qiao et al., 2019), given enough engineering geology data. For the study area, soil or rock strength parameters were unavailable for use in the deterministic method. We therefore used three probabilities (spatial, temporal, and size) to resolve questions on where and how potential landslides will occur with absolute magnitude in a given time (Guzzetti et al., 2005) using the following expression:

$$H = P(S) \times P(N_L) \times P(A_L), \tag{1}$$

where $H$ represents the hazard probability, $P(S)$ is the spatial probability, $P(N_L)$ is a temporal probability, and $P(A_L)$ is size probability.

#### 3.1.1 Spatial probability

Spatial probability, based on the concept of susceptibility, assesses the locations where a mass movement exists or may

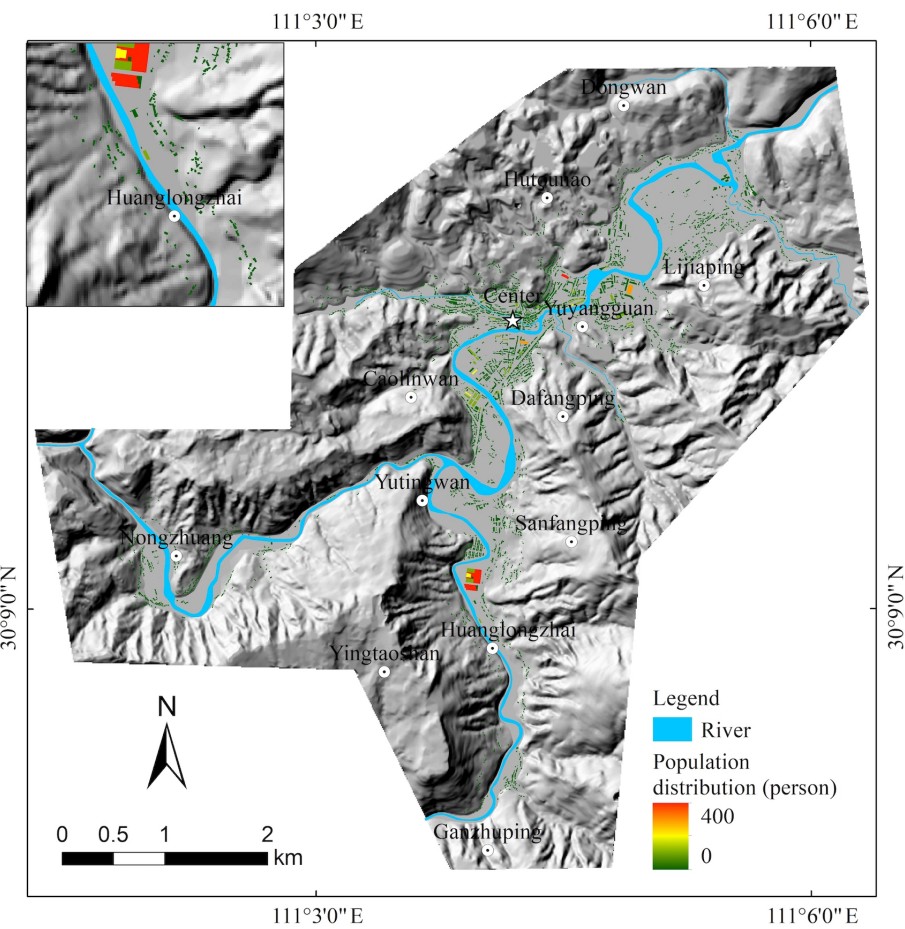

**Figure 6.** Population distribution map of the Yuyangguan community.

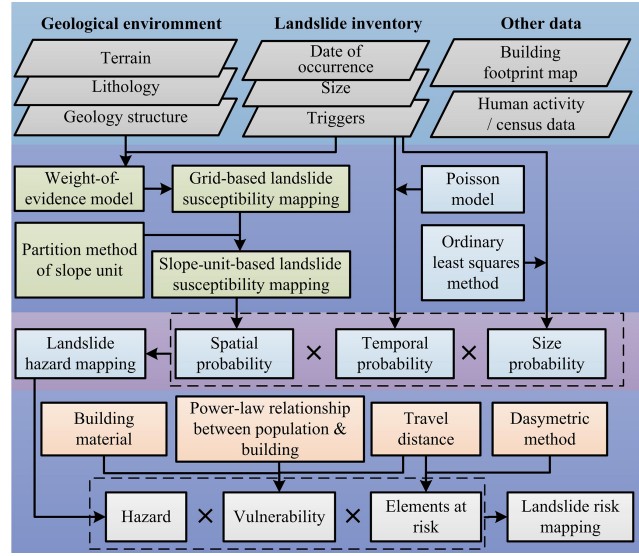

**Figure 7.** Flowchart of the methodology for landslide hazard risk assessment.

potentially occur. Landslide susceptibility mapping (LSM) is now widely used by researchers (Ayalew et al., 2004; Fell et al., 2008b; Van Westen et al., 2008; Guzzetti et al., 2012). In this study, morphometric and geoenvironmental factors, including altitude, slope, aspect, curvature, slope structure, distance to rivers, and proximity to roads, were chosen as variables. The morphometric factor maps were derived from the DEM with a $10\,\mathrm{m} \times 10\,\mathrm{m}$ resolution obtained by a UAV. Geoenvironmental factors, such as lithology and faults, were prepared and transformed from shapefiles to grid-based maps using a 1 : 50 000 scale geological field map from the China Geological Survey (http://www.cgs.gov.cn/, last access: September 2016). Detailed processing steps in ArcGIS are provided in Catani, Casagli, Ermini, Righini, and Menduni (2005). The commonly applied weight of evidence (WoE) method was used to assess landslide susceptibility in this study. This is a probabilistic model considering evidence factors of landslides, based on the conditional independence hypothesis (Hong et al., 2017). In the WoE method, $W^+$ and $W^-$ were used as the weights where the evidence was present or absent, respectively, and the contrasts (differ-

**Table 2.** Historical landslides investigated in the field by the authors.

| Date (month/year) | Landslide ID | Coordinates | | Material | Bedrock | Volume ($\times 10\,000\,\text{m}^3$) | Area ($\times 10\,000\,\text{m}^2$) | Triggering factors |
| --- | --- | --- | --- | --- | --- | --- | --- | --- |
| | | $X$ | $Y$ | | | | | |
| Seven landslides were located in the Yuyangguan community. | | | | | | | | |
| Jun 1981 | YYG02 | 506 615 | 333 5500 | Soil | $S_1 \text{lr}$ | 49.8 | 4 | R and SC |
| Jan 1989 | YYG05 | 505 788 | 3 338 880 | Soil | $S_2 s$ | 10.4 | 3.5 | R |
| Dec 2007 | YYG06 | 507 320 | 3 340 080 | Soil and rock | $O_1 d$ | 8 | 1.6 | R and SC |
| Jun 2010 | YYG07 | 507 381 | 3 341 680 | Rock | $O_1 d$ | 22.2 | 2.2 | R and SC |
| May 2013 | YYG04 | 506 275 | 3 338 610 | Soil | $S_2 s$ | 10.7 | 1.5 | R and SC |
| Jul 2013 | YYG01 | 506 692 | 3 334 830 | Soil | $S_2 s$ | 147.6 | 5.9 | R |
| Jan 2014 | YYG03 | 506 828 | 3 337 320 | Soil | $S_1 l$ | 20.9 | 1.7 | R and SC |
| Nineteen landslides were located in the Chengguan community. | | | | | | | | |
| Jun 1967 | CG08 | 468 193 | 3 341 670 | Soil | $O_2 b$ | 79.5 | 5.3 | R |
| Jul 1969 | CG09 | 468 331 | 3 341 980 | Soil and rock | $O_2 b$ | 16.8 | 2.4 | R |
| Jul 1969 | CG17 | 468 119 | 3 342 020 | Soil | $S_1 l$ | 4 | 0.5 | R |
| Jun 1991 | CG19 | 467 889 | 3 341 710 | Soil | $S_1 l$ | 4.32 | 0.54 | R |
| Jul 1991 | CG03 | 467 576 | 3 340 630 | Rock | $O_2 b$ | 1.92 | 0.48 | R |
| Jun 1992 | CG06 | 468 072 | 3 341 460 | Rock | $O_2 b$ | 34.5 | 2.3 | R |
| Jun 1992 | CG11 | 468 841 | 3 342 740 | Soil | $S_1 l$ | 7 | 0.5 | R |
| Jun 1992 | CG12 | 468 354 | 3 342 720 | Soil | $S_1 l$ | 9.6 | 1.2 | R |
| Jul 1994 | CG13 | 467 816 | 3 342 670 | Soil | $S_1 l$ | 4.55 | 0.65 | R |
| Jul 1994 | CG14 | 467 888 | 3 342 520 | Soil | $S1_l$ | 8.2 | 0.82 | R |
| Jul 1996 | CG16 | 468 188 | 3 342 070 | Soil | $S_1 l$ | 3.6 | 0.6 | R |
| Jul 1997 | CG07 | 468 295 | 3 341 510 | Soil | $O_2 b$ | 14.3 | 1.1 | R |
| Feb 1998 | CG15 | 467 956 | 3 342 230 | Soil | $S_1 l$ | 8 | 0.8 | R |
| Jul 2002 | CG05 | 467 830 | 3 340 990 | Soil | $O_2 b$ | 55.9 | 4.3 | R and SC |
| Aug 2005 | CG04 | 467 659 | 3 340 830 | Rock | $O_2 b$ | 5.04 | 0.56 | SC |
| Jul 2007 | CG18 | 467 707 | 3 341 990 | Soil | $S_1 l$ | 13.2 | 1.1 | R |
| May 2009 | CG01 | 466 943 | 3 339 730 | Soil | $S_1 l$ | 5.4 | 0.45 | R |
| Mar 2012 | CG10 | 468 597 | 3 342 480 | Soil | $O_2 b$ | 3.18 | 0.53 | R |
| Apr 2012 | CG02 | 467 038 | 3 339 980 | Soil | $S_1 l$ | 1.05 | 0.21 | R |

R – rainfall; SC – slope cut. The code of bedrock is listed in Table 1. The projected coordinate system is Xian_1980_3_Degree_GK_Zone_37 with projection of Gauss_Kruger.

ences between $W^+$ and $W^-$) were used as the weight for the morphometric and geoenvironmental factors.

The effectiveness of the LSM was tested using the receiver operating characteristics (ROC) curve (Metz, 1978; Zezere et al., 2017), with the area under the ROC curve used to assess the success rate.

The above grid-based LSM data were then converted into a slope-unit-based susceptibility map. The susceptibility value of each slope was calculated from the average value of the susceptibility of the grids within the slope. The slope unit was subdivided using the hydrology analysis method in the ArcGIS platform. The slope-unit-based spatial probability map was classified into five classes, including very high, high, moderate, low, and very low.

### 3.1.2 Temporal probability

Landslide temporal probability $P(N_\text{L})$ is evaluated by assuming that slope failures are independent random point events in the time domain (Crovelli, 2000; Guzzetti et al., 2006). In this study, the Poisson model (Crovelli, 2000) was adopted for constructing temporal probability. It is the exceedance probability of landslide occurrence during a given period (see Eq. 2), meaning the probability of experiencing one or more landslides during a given time, and derived as

$$P(N_\text{L}) = 1 - e^{-T/\text{RI}}, \text{RI} = t/N, \tag{2}$$

where $T$ is the return period, e.g., 1, 10, 20, and 50 years; the recurrence interval (RI) is the historical mean recurrence interval for each slope unit; $t$ is the temporal interval of the landslide database; and $N$ is the number of landslides recorded in each slope. If the historical landslide database is incomplete, $N$ on a slope unit with a very high susceptibility class in the LSM is assigned as 1.

### 3.1.3 Size probability

Landslide size probability is calculated based on the relationship between landslide volume and cumulative frequency. Guzzetti et al. (2005) used the probability density function of the landslide area to predict the probability of a specific landslide area in each slope unit. Stark and Hovius (2001) found that landslides in New Zealand and Taiwan fitted with a double Pareto probability distribution, and two differences in our study will improve the application. The first is that the present study seeks the most suitable distribution for the case study area. Therefore, we compared the distributions stated previously and introduced another type using the ordinary least squares (OLS) method in the MATLAB software, with the best fit used for probability calculation. The second is that landslide volume is an acceptable indicator for risk control practice in the study area. Therefore, we converted the landslide size probability distribution from area to volume using the volume–area relationship simulated by the OLS method. For this, we used the historical landslides in the database of the study area in Table 2. The hazard database of the Chengguan community was implemented because of the landslide records limitation, but the geoenvironment and hazards in both communities are similar. Meanwhile, the two size scenarios were determined from the distribution of landslide volume in the case study area.

### 3.2 Data preparation for elements at risk

The next step was determining the elements at landslide risk. This study focuses on residential buildings and people within. The building footprint map (see Fig. 5) was interpreted from the 2013 UAV image data. The building structure (reinforced, reinforced concrete, masonry, and wooden) and numbers of floors are involved in the building map database. To express the risk in monetary values, we used the economic value of the buildings (see Table 3) obtained from the Department of Natural Resources of Hubei Province. These data were converted to building values by multiplying the unit economic values, footprint areas, and the number of floors. Data for the population in buildings were obtained by integrating the information from the China population census data (2010) and the sampling survey. The average number of people per building was calculated by applying a dasymetric mapping approach (a methodology for generating a surface-based representation of the population; Mennis, 2003), which contributes to the data of the population in each building. To assess the element at risk, the building footprint map was then combined with the potential landslide influence area at the community level. The influence area of each slope was semiquantitatively determined by calculating the travel distances using the following formula (Hungr et al., 2005):

$$\log(H/L) = A + B \times \log V, \tag{3}$$

Table 3. The economic value of buildings in the Yuyangguan community (the Department of Natural Resources of Hubei Province, 2016).

| Typology | USD 100 per m$^2$ | Number of floors |
|---|---|---|
| Reinforced | 21.84 | 20–32 |
| Reinforced concrete | 9.31 | 6–20 |
| Masonry | 5.60 | 2–6 |
| Wooden | 0.70 | 1–2 |

where $L$ is the travel distance, $H$ is the slope height, $V$ is slope volume, and $A$ and $B$ are constants. These constants are referred to in Corominas (1996).

### 3.3 Vulnerability analysis and risk assessment

Quantitative vulnerability analysis is still a challenge in landslide risk assessment (Chen et al., 2011; Peduto et al., 2017). Physical vulnerability assessment is performed in a large- or local-scale area (Fell et al., 2008b; Li et al., 2010; Quan Luna et al., 2011). In this study, the physical vulnerability was semiquantitatively determined for buildings, using two indicators. The first indicator relates to building structures, such as reinforced, reinforced concrete, masonry, and wooden, while the second indicator is the landslide travel distance. The assumption is that vulnerability is 1.0 for the buildings on the slope, and it decreases from the toe of the surface rupture to the farthest travel distance. We therefore propose vulnerability values for different types of buildings (see Table 4).

The vulnerability of the population in buildings follows a power–law relationship with building vulnerability (Li et al., 2010) as follows (see Eq. 4):

$$V_p = 0.0014 \times e^{6.07 \times V_b}, \tag{4}$$

where $V_p$ is the vulnerability of the population in buildings and $V_b$ is the vulnerability of a building.

A landslide risk map was then generated in ArcGIS based on the concept defined by the IAEG (International Association of Engineering Geology) and Varnes (1984) as "the expected number of lives lost, persons injured, damage to property, and disruption of economic activity due to a particularly damaging phenomenon for a given area and reference period". The conceptual equation for risk is

$$R = H \times V \times E, \tag{5}$$

where $R$ is the expected loss for some return period, and $H$ is the landslide probability of some return period with a given size scenario. In the present study, $V$ is the physical vulnerability of buildings or the population in the buildings, and $E$ is the quantification of the exposed elements at risk. Using Eq. (5), the risk curve is fitted by plotting the probability

**Table 4.** Vulnerability value of buildings impacted by landslides (proposed by authors).

| | Influence area | | | | |
|---|---|---|---|---|---|
| | Zone 1 | Zone 2 | Zone 3 | Zone 4 | |
| Reinforced | 0.1 | 0.3 | 0.5 | 1 | |
| Reinforced concrete | 0.2 | 0.5 | 0.7 | 1 | |
| Masonry | 0.3 | 0.7 | 0.9 | 1 | |
| Wooden | 0.4 | 0.9 | 1 | 1 | |

$L$ is travel distance; $H$ is slope height.

versus potential loss, with the annual risk calculated from the area under the risk curve (Van Westen et al., 2002).

## 4  Results

This section provides the results of the case study to illustrate the application of the proposed framework and methodology in Sect. 3.

### 4.1  Landslide susceptibility assessment

In assessing landslide susceptibility, we investigated the elevation, slope, aspect, curvature, lithology, slope structure, distance to fault, rivers, and roads. The weights and contrasts values from the WoE method for LSM of Yuyangguan are presented in Table 5. According to the contrast values explained in Sect. 3.1 for lithology, the Ordovician limestones and shales ($O_2g$) and Quaternary eluvium ($Q_4^{d_l+e_l}$) are the top two units, implying that these units are susceptible to erosion. Under this background, heavy rainfall in the area accelerates erosion, thus triggering landslides.

General road construction is of secondary importance, with a contrast value of 0.95. Data in Table 5 also reveal generally significant morphometric factors in the study area. For aspect and elevation, the contrast values are elevated on the north-facing slopes with an elevation from 0 to 260 m a.s.l., but low on the south-facing slopes with an elevation above 350 m a.s.l. For the slope, the steeper the slope is, the higher the landslide probability is, with the contrast value for slopes ranging from 10 to 30° being 0.19, indicating a relatively high landslide probability.

The grid-based susceptibility map was converted to a slope-unit-based map with 701 slope units in total. The slope-unit-based susceptibility map was ordered into five classes, ranging from very low to very high (Fig. 8). The performance from the map reveals an accuracy of 84 %, using the ROC curve. The landslide susceptibility is very high for the north-facing slopes along the main road, especially where $Q_4^{d_l+e_l}$ and $O_2g$ rocks are present. These results correspond well with the contrast values presented in Table 5.

### 4.2  Landslide hazard probability

The landslide hazard involves spatial, temporal, and size probabilities. The landslide data for Yuyangguan presented in Table 2 cover the 33 years from 1981 to 2013. For each slope unit, the historical mean RI is calculated using Eq. (2). Assuming that the past is an indicator of the future, landslides in the study area may be modeled over the next 50 years, based on the past 50 years.

Four landslide temporal probability maps are displayed in Fig. 9 for four return periods (5, 10, 20, and 50 years). The map for 50 years, for example, shows the highest probability for the slope units experiencing landslide events among the four maps. Slope units with high and very high probability values ($> 0.5$) cluster on the first slope zones around the community. The probability values on the slopes increase from the return period of 5 years. For example, the enlarged windows in maps of Fig. 9 demonstrate that the slope in Huanglongzhai village experienced a very low probability in the 5 years to a high class in the 50-year return period.

The landslide probability distribution curves are created using three different fitting functions, as displayed in Fig. 10. In comparison to the inverse gamma and the power law distributions, the function by the OLS method shows the best fit, with the lowest root mean square error of 0.05. This indicates that the landslide frequency distribution function by the OLS method is the most appropriate technique to apply in Yuyangguan. The volume–area relationship is analyzed in Fig. 11a with an $R^2$ value of 0.915. This indicates that converting the size probability distribution from landslide area data to volume is feasible in the study area. Meanwhile, two size scenarios are determined from the cumulative frequency curve (Fig. 11b) based on a landslide volume of 50 000 and 100 000 m$^3$. The number of landslides begins to increase rapidly in Fig. 11b when the volume is greater than 50 000 m$^3$, representing a probable threshold value of landslide hazard volume, while a landside volume greater than 100 000 m$^3$ represents the maximum for the landslides, as shown in Fig. 11b. The value of 100 000 m$^3$ is a standard threshold for landslide classification provided by the China Geological Survey and a widely accepted value in landslide hazard risk control in China.

**Table 5.** The weight and contrast values by the weight-of-evidence model for landslide susceptibility mapping of Yuyangguan, western Hubei, China.

| Factors | Classes | Area of domain % | Landslide area % | $W^+$ | $W^-$ | $C = W^+ - W^-$ | $S^2(W^+)$[a] | $S^2(W^-)$[b] | $S(C)$[c] | $C/S(C)$[d] |
|---|---|---|---|---|---|---|---|---|---|---|
| Slope | 0–10 | 11.03 | 3.66 | −1.11 | 0.08 | −1.19 | 0.01 | 0.00 | 11.04 | −13.15 |
| (degrees) | 10–30 | 55.76 | 67.48 | 0.19 | −0.31 | 0.5 | 0.00 | 0.00 | 27.44 | 13.84 |
| | 30–50 | 30.77 | 28.86 | −0.07 | 0.03 | −0.09 | 0.00 | 0.00 | 26.53 | −2.45 |
| | 50–90 | 2.44 | 0 | 0 | 0.02 | −0.02 | – | 0.00 | – | – |
| Elevation | 0–260 | 30.99 | 51.21 | 0.51 | −0.35 | 0.86 | 0.00 | 0.00 | 29.24 | 25.15 |
| (meters) | 260–350 | 35.75 | 43.11 | 0.19 | −0.12 | 0.31 | 0.00 | 0.00 | 28.99 | 9.07 |
| | 350–440 | 19.81 | 5.68 | −1.26 | 0.16 | −1.42 | 0.01 | 0.00 | 13.60 | −19.36 |
| | 440–680 | 13.44 | 0 | 0 | 0.15 | −0.15 | – | 0.00 | – | – |
| Aspect | Plan area | 2.32 | 0.14 | −2.79 | 0.02 | −2.81 | 0.20 | 0.00 | 2.23 | −6.29 |
| (degrees) | NNE (0–45) | 12.08 | 10.06 | −0.19 | 0.02 | −0.21 | 0.00 | 0.00 | 17.63 | 0.00 |
| | NEE (45–90) | 10.86 | 17.23 | 0.47 | −0.08 | 0.54 | 0.00 | 0.00 | 22.05 | 12.00 |
| | SEE (90–135) | 11.48 | 7.06 | −0.49 | 0.05 | −0.54 | 0.00 | 0.00 | 15.03 | −8.11 |
| | SSE (135–180) | 12.51 | 7.44 | −0.52 | 0.06 | −0.58 | 0.00 | 0.00 | 15.40 | −8.96 |
| | SSW (180–225) | 12.25 | 7.2 | −0.54 | 0.06 | −0.59 | 0.00 | 0.00 | 15.17 | −8.99 |
| | SWW (225–270) | 12.16 | 5.88 | −0.73 | 0.07 | −0.8 | 0.00 | 0.00 | 13.81 | −11.09 |
| | NWW (270–315) | 12.18 | 15.59 | 0.25 | −0.04 | 0.29 | 0.00 | 0.00 | 21.21 | 6.17 |
| | NNW (315–360) | 14.17 | 29.39 | 0.74 | −0.2 | 0.94 | 0.00 | 0.00 | 26.56 | 24.97 |
| Lithology | $Q^{al+pl}$ | 0.78 | 0.06 | −2.61 | 0.01 | −2.62 | 0.50 | 0.00 | 1.41 | −3.70 |
| | $Q^{dl+el}$ | 8.08 | 50.95 | 1.91 | −0.63 | 2.54 | 0.00 | 0.00 | 28.83 | 73.28 |
| | $P_1$mn | 0.58 | 0 | 0 | 0.01 | −0.01 | – | 0.00 | – | – |
| | $P_1q$ | 0.75 | 0 | 0 | 0.01 | −0.01 | – | 0.00 | – | – |
| | $C_2$hn | 0.26 | 0 | 0 | 0 | 0 | – | 0.00 | – | – |
| | $D_3x$ | 0.63 | 0 | 0 | 0.01 | −0.01 | – | 0.00 | – | – |
| | $D_3h$ | 1.24 | 0 | 0 | 0.01 | −0.01 | – | 0.00 | – | – |
| | $D_2y$ | 7.16 | 0 | 0 | 0.08 | −0.08 | – | 0.00 | – | – |
| | $S_2s$ | 22.07 | 27.61 | 0.23 | −0.07 | 0.3 | 0.00 | 0.00 | 26.15 | 7.88 |
| | $S_1$lr | 21.34 | 11.99 | −0.58 | 0.11 | −0.7 | 0.00 | 0.00 | 19.06 | −13.26 |
| | $S_1l$ | 15.18 | 0 | 0 | 0.17 | −0.17 | – | 0.00 | – | – |
| | $O_2b$ | 4.54 | 0 | 0 | 0.05 | −0.05 | – | 0.00 | – | – |
| | $O_2g$ | 3.33 | 9.37 | 1.06 | −0.07 | 1.12 | 0.00 | 0.00 | 16.89 | 18.94 |
| | $O_1d$ | 3.83 | 0.03 | −4.9 | 0.04 | −4.94 | 1.00 | 0.00 | 1.00 | −4.94 |
| | $O_1h$ | 4.71 | 0 | 0 | 0.05 | −0.05 | – | 0.00 | – | – |
| | $O_1n$ | 4.69 | 0 | 0 | 0.05 | −0.05 | – | 0.00 | – | – |
| | $\epsilon_2sn$ | 0.84 | 0 | 0 | 0.01 | −0.01 | – | 0.00 | – | – |
| Curvature | Concave | 48.71 | 51.94 | 0.07 | −0.07 | 0.14 | 0.00 | 0.00 | 29.25 | 4.11 |
| | Straight/flat | 0.87 | 0.03 | −3.42 | 0.01 | −3.43 | 1.00 | 0.00 | 1.00 | −3.43 |
| | Convex | 50.42 | 48.03 | −0.05 | 0.04 | −0.09 | 0.00 | 0.00 | 29.26 | −2.54 |
| Slope | Overdip | 9.88 | 12.28 | 0.22 | −0.03 | 0.25 | 0.00 | 0.00 | 19.19 | 4.78 |
| structure | Underdip | 5.8 | 2.39 | −0.89 | 0.04 | −0.93 | 0.01 | 0.00 | 8.98 | −8.32 |
| | Oblique | 15.19 | 6.71 | −0.82 | 0.1 | −0.92 | 0.00 | 0.00 | 14.70 | −13.49 |
| | Transverse | 32.32 | 33.29 | 0.03 | −0.02 | 0.05 | 0.00 | 0.00 | 27.59 | 0.00 |
| | Anaclinal | 36.82 | 45.33 | 0.21 | −0.15 | 0.36 | 0.00 | 0.00 | 29.14 | 10.47 |
| Distance to faults | > 100 | 84.96 | 88.47 | 0.04 | −0.27 | 0.31 | 0.00 | 0.00 | 18.72 | 5.80 |
| (meters) | < 100 | 15.04 | 11.53 | −0.27 | 0.04 | −0.31 | 0.00 | 0.00 | 18.72 | −5.80 |
| Distance to rivers | > 50 | 89.57 | 87.26 | −0.03 | 0.2 | −0.23 | 0.00 | 0.00 | 19.50 | −4.46 |
| (meters) | < 50 | 10.43 | 12.74 | 0.2 | −0.03 | 0.23 | 0.00 | 0.00 | 19.50 | 4.46 |
| Distance to roads | Outside buffer | 83.76 | 69.63 | −0.19 | 0.64 | −0.82 | 0.00 | 0.00 | 26.84 | −22.09 |
| | General road (< 100) | 9.73 | 21.59 | 0.81 | −0.14 | 0.95 | 0.00 | 0.00 | 23.95 | 22.85 |
| (meters) | Highroad (< 200) | 6.52 | 8.79 | 0.3 | −0.02 | 0.33 | 0.00 | 0.00 | 16.55 | 5.44 |

The total area of the community is 29.14 km$^2$. Total landslide area is 3740 pixels. [a] Variance of $W^+$. [b] Variance of $W^-$. [c] Standard deviation of contrast. [d] Studentized value of contrast.

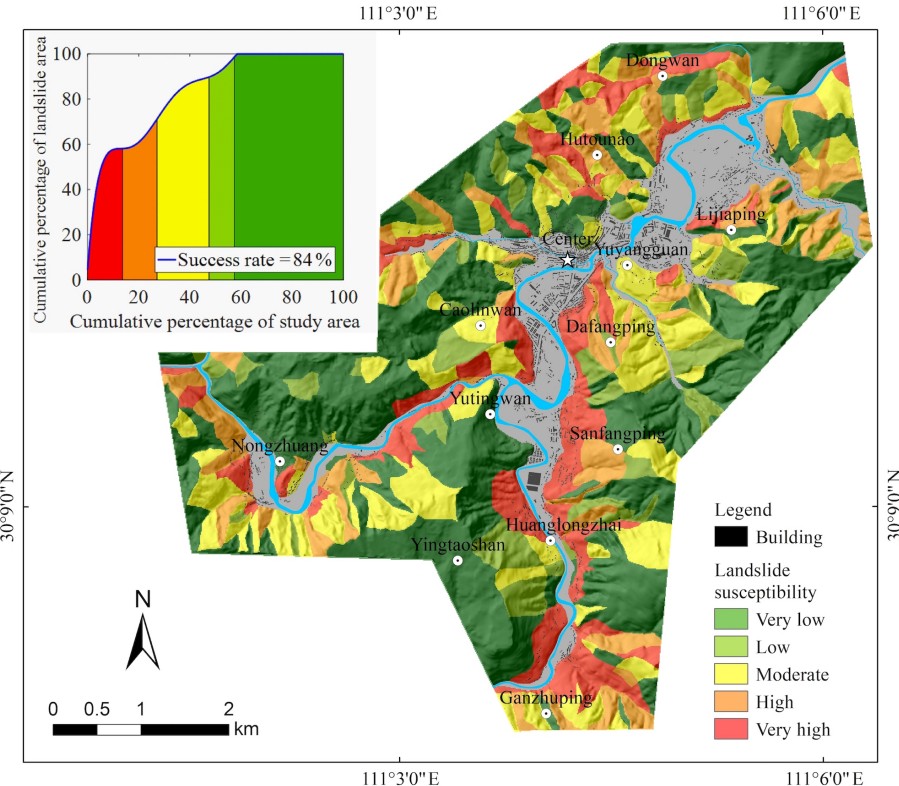

**Figure 8.** Slope-unit-based LSM at the community level, converted from the grid-based LSM using the weight-of-evidence method for Yuyangguan, western Hubei, China.

Integrating the spatial probability in Fig. 9, temporal probability in Fig. 10, and the magnitude–frequency relationships in Fig. 11, eight hazard maps were produced using Eq. (1). Through these maps, we demonstrate the hazard probability for each slope unit for four return periods (5, 10, 20, and 50 years) and landslide sizes equal to or greater than $50\,000\,\mathrm{m^3}$ and equal to or greater than $100\,000\,\mathrm{m^3}$. For example, Fig. 12 shows the four landslide hazard maps for the four return periods and the landslide size scenarios equal to or greater than $50\,000\,\mathrm{m^3}$. In these maps, the landslide hazard probability values comprise five categories from very low (0.0–0.1) to very high (0.4–0.5). The maps for the 50-year return period show very high hazard probabilities for most building areas in the community at the bottom or near slopes. Conversely, buildings located in the community center are in areas from low to very low hazard probability class.

### 4.3 Element-at-risk and vulnerability assessment

We assume that the buildings and population in the buildings are exposed to the slopes with high and very high class probability in the hazard maps. Therefore, the element-at-risk maps for buildings and the population generated from the data in Sect. 2 and methodology in Sect. 3.2 are consistent with the four return periods and two size scenarios of the landslide hazard probability maps.

Data in Table 6 show no exposure for the return periods of 5 years and 10 years, while for the 50-year return period and $50\,000\,\mathrm{m^3}$ size scenario, $570\,000\,\mathrm{m^2}$ of hosing areas and 14 257 persons are exposed to landslide risk. The data also highlight the potential for damaged building areas and the number of persons in the buildings for return periods of 20 to 50 years. The vulnerability value is indirectly calculated by Eq. (4) in Sect. 3.3. In the size scenario of landslides for the volume equal to or greater than $50\,000\,\mathrm{m^3}$, a sharp increase in exposure exists for built areas and their population, with the exposure more than doubling from 20 to 50 years. Similar results emerge for the size scenario of landslide volume equal to or greater than $100\,000\,\mathrm{m^3}$. Comparing the two size scenarios, we find that the number exposed for the $100\,000\,\mathrm{m^3}$ volume is lower than that for $50\,000\,\mathrm{m^3}$. The percentage of building areas exposed to landslides in the 20-year return period are 8.76 % for the $50\,000\,\mathrm{m^3}$ volume, whereas it is 5.86 % for the $100\,000\,\mathrm{m^3}$ volume, probably due to the lower hazard probability in the latter scenario. A similar tendency is observed for the population in buildings, with about 30 % exposed to landslides for the 50-year return period.

Accordingly, eight vulnerability maps for buildings and eight vulnerability maps for the population in the buildings are created for the four return periods and two size scenarios. For example, Figs. 13 and 14 show the resulting vulnerabil-

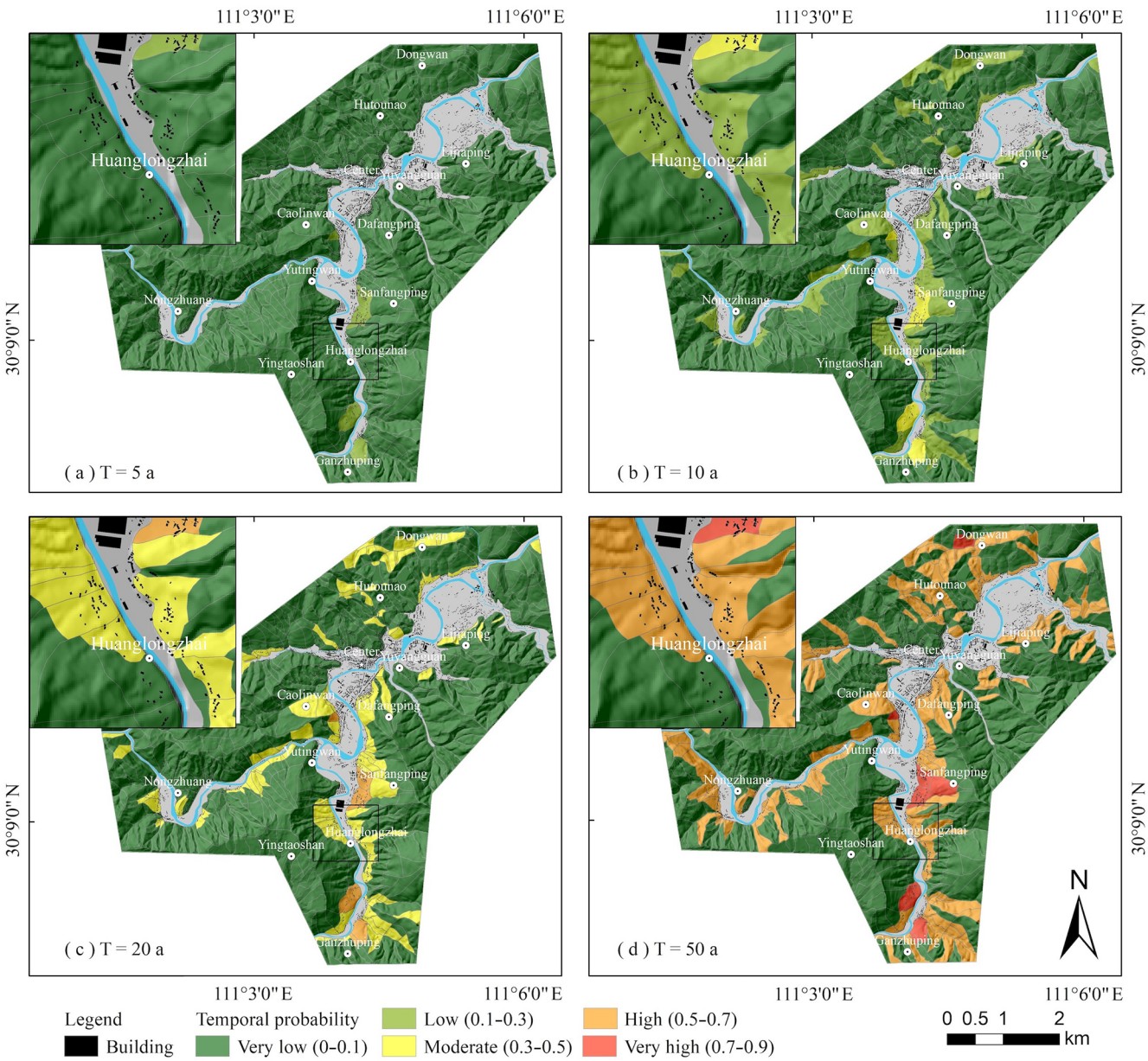

**Figure 9.** Landslide temporal probability maps based on the Poisson model, showing the exceedance probability of landslide occurrence in each slope unit for four return periods (5, 10, 20, 50 years).

ity maps for buildings and the population in buildings for the four return periods and when the landslide volume is equal to or greater than 50 000 m$^3$. Most exposed buildings are located on slopes with high or very high hazard probability, including the Caolinwan, Dafangping, and Sanfangping villages. The buildings and population in buildings in these central communities are not exposed to slope hazard because of the very low class (0–0.1) of landslide probability.

As presented in Table 4 in Sect. 3.3, we assigned a vulnerability value of 1.0 to buildings on slopes. This causes buildings outside the flat areas in the community to display very high vulnerability values in Fig. 13, involving five cat-

egories ranging from very low to very high. For the 50-year return period and 50 000 m$^3$ size scenario, 18 % of built areas exhibit very high class vulnerability. The vulnerability value for the population was then assigned according to Eq. (4) in Sect. 3.3 from the building vulnerability result. The very high class population vulnerability shown in Fig. 14 was over 0.5, with the area representing about 10 % of the people exposed, concentrated in the slope influence areas of Dafangping, Caolinwan, and Sanfangping or on the slopes with the very high class hazard probabilities displayed in Fig. 12.

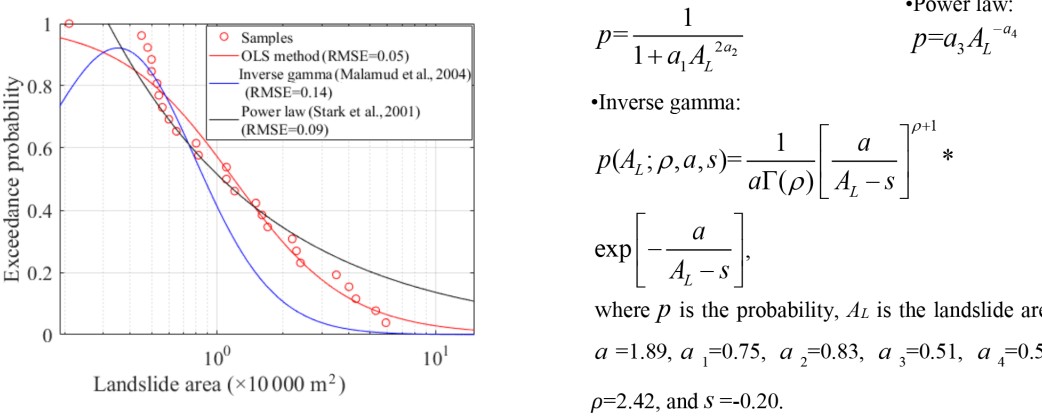

•OLS method:
$$p=\frac{1}{1+a_1 A_L^{2a_2}}$$

•Power law:
$$p=a_3 A_L^{-a_4}$$

•Inverse gamma:
$$p(A_L;\rho,a,s)=\frac{1}{a\Gamma(\rho)}\left[\frac{a}{A_L-s}\right]^{\rho+1}*$$
$$\exp\left[-\frac{a}{A_L-s}\right],$$

where $p$ is the probability, $A_L$ is the landslide area, $a=1.89$, $a_1=0.75$, $a_2=0.83$, $a_3=0.51$, $a_4=0.58$, $\rho=2.42$, and $s=-0.20$.

**Figure 10.** Magnitude–frequency relationships simulated by the ordinary least squares (OLS) method, inverse gamma, and power law for landslide in western Hubei. The samples are presented in Table 2 and involve the Chengguan database except for the study area.

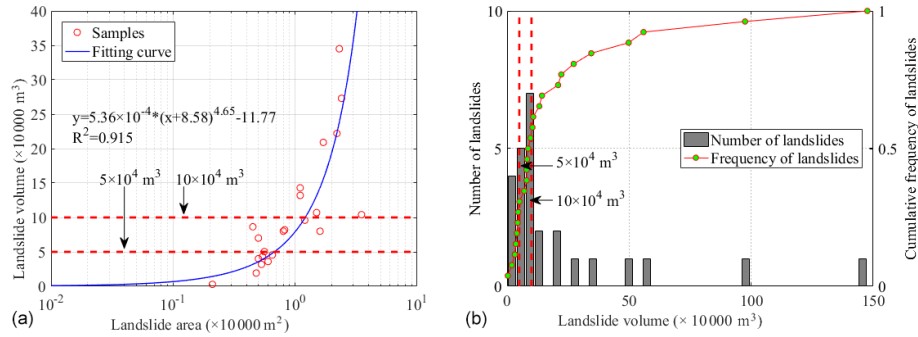

**Figure 11. (a)** Relationship between landslide volume and area ($R^2 = 0.915$) used to convert magnitude–frequency relationships for the area to landslide volume. **(b)** Frequency distribution of landslide volume. The samples are presented in Table 2, involving the Chengguan database, except for the study area. CE2

## 4.4 Risk assessment

Landslide risk maps are then generated from the eight hazard maps, eight vulnerability maps, and the values of the elements at risk by Eq. (5). An example of a risk map for the population loss is shown in Fig. 15. The map for the 50-year return period, for example (see Fig. 15d), shows that potential loss is concentrated in the urban and densely populated areas along the Yuyangguan River, especially at the toe of the slope of the Caolinwan, Dafangping, and Sanfangping villages.

The final population and building risks for four return periods and two size scenarios are presented in Table 6. Accordingly, no potential losses emerge for the 5- and 10-year return periods. The absence of risk for the 5-year return period is proven by the fact that no casualties or economic losses have been reported in the Yuyangguan community since 2014. For the volume equal to or greater than $50\,000\,\mathrm{m}^3$, however, the potential casualties are 771 persons, representing 1.68 % of the total population in the community. The economic losses are estimated at USD 8.57 million for the 20-

year return period, amounting to 3.7 % of economic values associated with buildings. As presented in Table 6, the risk is expected to double in the next 50-year return period. For the volume equal to or greater than $100\,000\,\mathrm{m}^3$, the potential loss is lower, with 1.27 % of the population exposed to landslides, amounting to a decrease of 0.41 % relative to the size scenario of $50\,000\,\mathrm{m}^3$.

## 5 Discussion

### 5.1 Discussion on landslide susceptibility map

The landslide susceptibility results demonstrate that lithology is the most important controlling factor. $Q_4^{dl+el}$, weathered Guniutan $O_2g$ limestone, and weathered $S_2s$ sandstone exhibit the top three weight values in lithology contribution in the susceptibility map. This is consistent with findings inform the fieldwork. The residual deposit and eluvium are composed of clay and gravels that are characterized by low strength and cover the main areas with intensive human activities in the Yuyangguan community. For the $O_2g$ bedrock, its

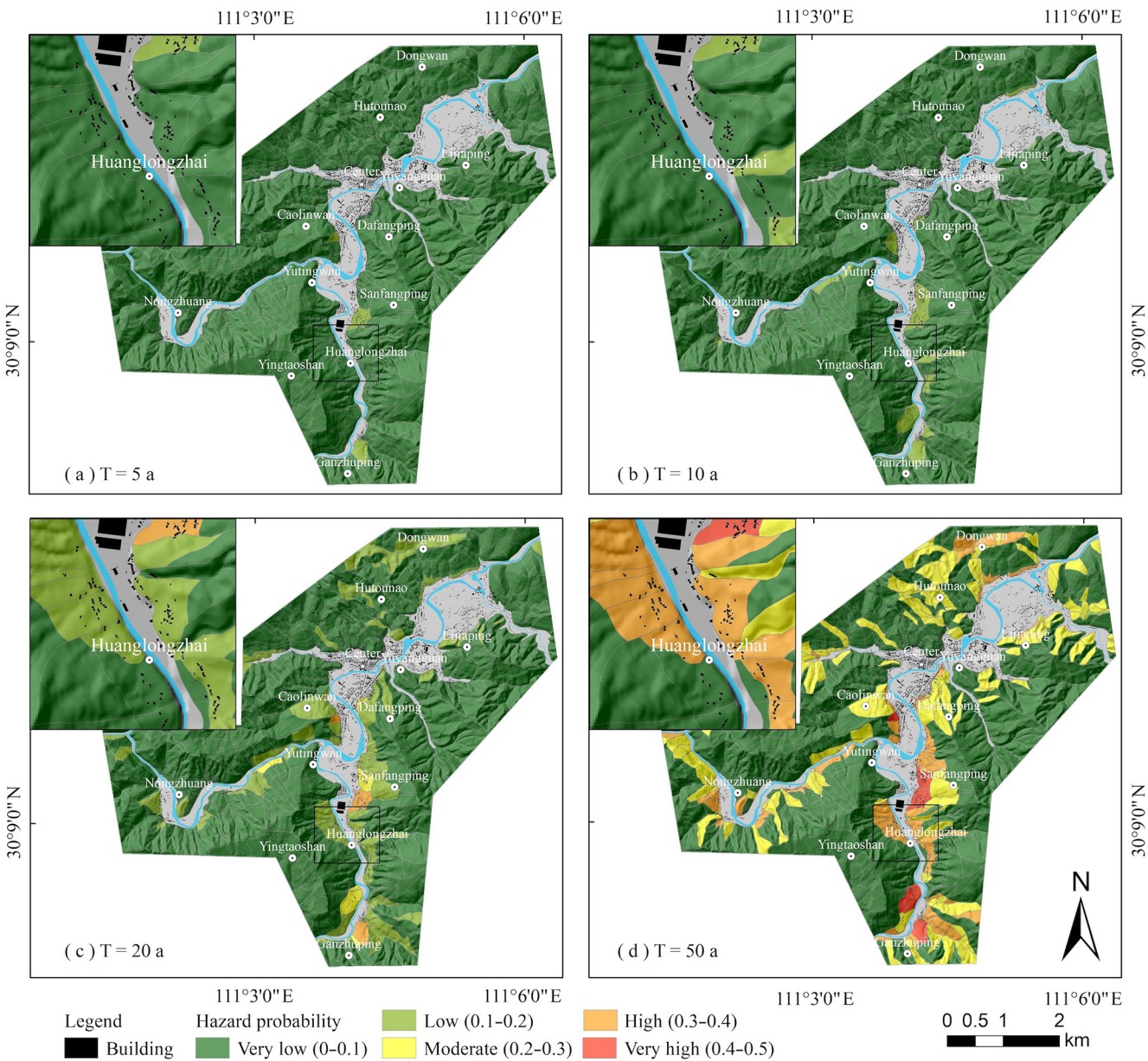

**Figure 12.** Landslide hazard maps for four return periods (5, 10, 20, and 50 years) and the landslide sizes scenario equal to or greater than $50\,000\,\text{m}^3$. The maps were generated by integrating the spatial probability in Fig. 8, temporal probability in Fig. 9, and the magnitude–frequency relationships in Fig. 10.

high strength makes it less vulnerable to a landslide. However, in the field, we found two groups of surface joints in the $O_2g$, and this helps to explain why the YYG07 landslide occurred (see Table 2). Field observations also reveal that most landslides occurred in $S_2s$ sandstones and shales. However, the weight value (0.3) of $S$ is lower than those of $Q_4^{d_l+e_l}$ and $O_2g$. This is because the area covered by the $S_2s$ formation is much larger than the area of $Q_4^{d_l+e_l}$ and $O_2g$ (see Fig. 1). Therefore, high landslide probability for slopes in $O_2g$ limestone must be considered, and attention should be

devoted to slopes where historical landslides are not as frequent now.

## 5.2 Discussion on landslide hazard assessment

Landslide hazard maps are generated for four return periods (5, 10, 20, and 50 years) and two size scenarios. Theoretically, the definition of hazard scenarios, as stated by L. Chen et al. (2016), should be based on the analysis of landslide occurrences and triggering events. Due to incomplete information on landslide dates in the community, it is challenging to

**Table 6.** Exposure and risk of buildings and population for landslides under four return periods (5, 10, 20, and 50 years) and two size scenarios (landslide volume equal to or greater than 50 000 or 100 000 m$^3$) by using the methodology in Sect. 3.2 and 3.3. (Number in brackets is in percentage.)

| | Size scenario<br>Return period (years) | Landslide volume equal to or greater<br>than 50 000 m$^3$ | | | | Landslide volume equal to or greater<br>than 100 000 m$^3$ | | | |
|---|---|---|---|---|---|---|---|---|---|
| | | 5 | 10 | 20 | 50 | 5 | 10 | 20 | 50 |
| Exposure | Population | 0 | 0 | 4073 (8.87 %) | 14 257 (31.05 %) | 0 | 0 | 2724 (5.93 %) | 13 746 (29.94 %) |
| | Building area (×10 000 m$^3$) | 0 | 0 | 16.3 (8.76 %) | 57 (30.64 %) | 0 | 0 | 10.9 (5.86 %) | 55 (29.57 %) |
| Risk | Casualties (person) | 0 | 0 | 771 (1.68 %) | 1485 (3.23 %) | 0 | 0 | 584 (1.27 %) | 1235 (2.69 %) |
| | Economic losses (× USD 1000) | 0 | 0 | 8627 (3.69 %) | 16 613 (7.10 %) | 0 | 0 | 6593 (2.82 %) | 13 360 (5.71 %) |
| Annual risk | Casualties (person) | | | 59 (0.13 %) | | | | 47 (0.10 %) | |
| | Economic losses (× USD 1000) | | | 661 (0.3 %) | | | | 513 (0.23 %) | |

establish the relationship between the landslide return period and triggering factor (rainfall in this case study). However, we observed that no landslide has occurred in the Yuyangguan community since 2014, which is consistent with the hazard result for the 5-year return period (see Fig. 12a). This means that the temporal probability approach using the Poisson model is suitable for landslide hazard assessment when the landslide database is missing the occurrence date and triggering event (e.g., rainfall) data.

In the size probability analysis, the landslide probability distribution is the key for quantifications. We found that the classical distribution model (Stark and Hovius, 2001; Malamud et al., 2004) failed to produce an excellent fitting performance in this study. The difference in the landslide size between Malamud's and our landslide database accounts for this inconsistency. No small landslides (< 1000 m$^3$ in Malamud's research) are present or recorded in the Yuyangguan community, and the simulation equation (in Fig. 10) is suitable for landslide risk assessment in this study. In the future, however, comparison with classical models should be undertaken, and other factors, such as triggers and landslide types, should be considered using a complete database or one with more landslide events.

## 5.3 Other limitations in risk results

Uncertainties exist in the final risk maps due to some other factors, such as element-at-risk data and their vulnerability or resilience. In this study, the data for buildings and population in buildings at risk are derived from empirical calculations for the landslide influence area. Further studies considering landslide material, pore water pressure, and ground surface characteristics besides slope height and volume used in this paper will be conducted by numerical modeling.

Meanwhile, the risk results cannot be tested because of a lack of historical damage data in the area, which is a common difficulty in China and other areas (Ghosh et al., 2011; L. Chen et al., 2016). Damage data are also crucial for vulnerability analysis. We considered building typology and location from the landslide to assess the physical vulnerability

based on local experts' opinions. However, the resilience of the element at risk contributes to a reasonable decrease in vulnerability. The mobility of persons and their characteristics (e.g., age, education, and physical disability) and disaster prevention capability of the government are not considered in the community. In the future, physical vulnerability curves for buildings and the population will be constructed for the area.

From susceptibility to risk assessment, in general, we assumed that landslides will occur under the same condition as historical landslides. However, morphometric and geoenvironmental conditions will change with time. For example, the slope degree, elements at risk, and land use cover may change because of new infrastructures, or the number of historical landslides may increase due to heavier rainfall. Future studies will be necessary, considering the changing conditions for more accurate results and more practical applications. So far, the series of maps represent a basis for landslide risk control and land use planning in the Yuyangguan community. Risk control measurements can be planned on each slope unit using a matrix combining landslide hazard probability and risk maps. For example, risk management on slope units with very high class hazard probability and very high risk can be suggested as requiring relocation or engineering works. The precondition is that a more detailed geotechnical investigation and comprehensive analysis be complemented. This achievement can be used by professionals on engineering geology. The slopes with high class hazard probability and midclass risk are suggested for monitoring, considering the importance of safety for the surrounding population, roads, or other elements at risk. Meanwhile, the results of the four return periods and two size scenarios are useful for multitemporal land use planning, including short term (5 years), mid-term (10–20 years), and long term (50 years) strategies. The annual risk value on each slope is useful for the cost–benefit analysis of risk decisions. This kind of achievements can be applied for government decision makers.

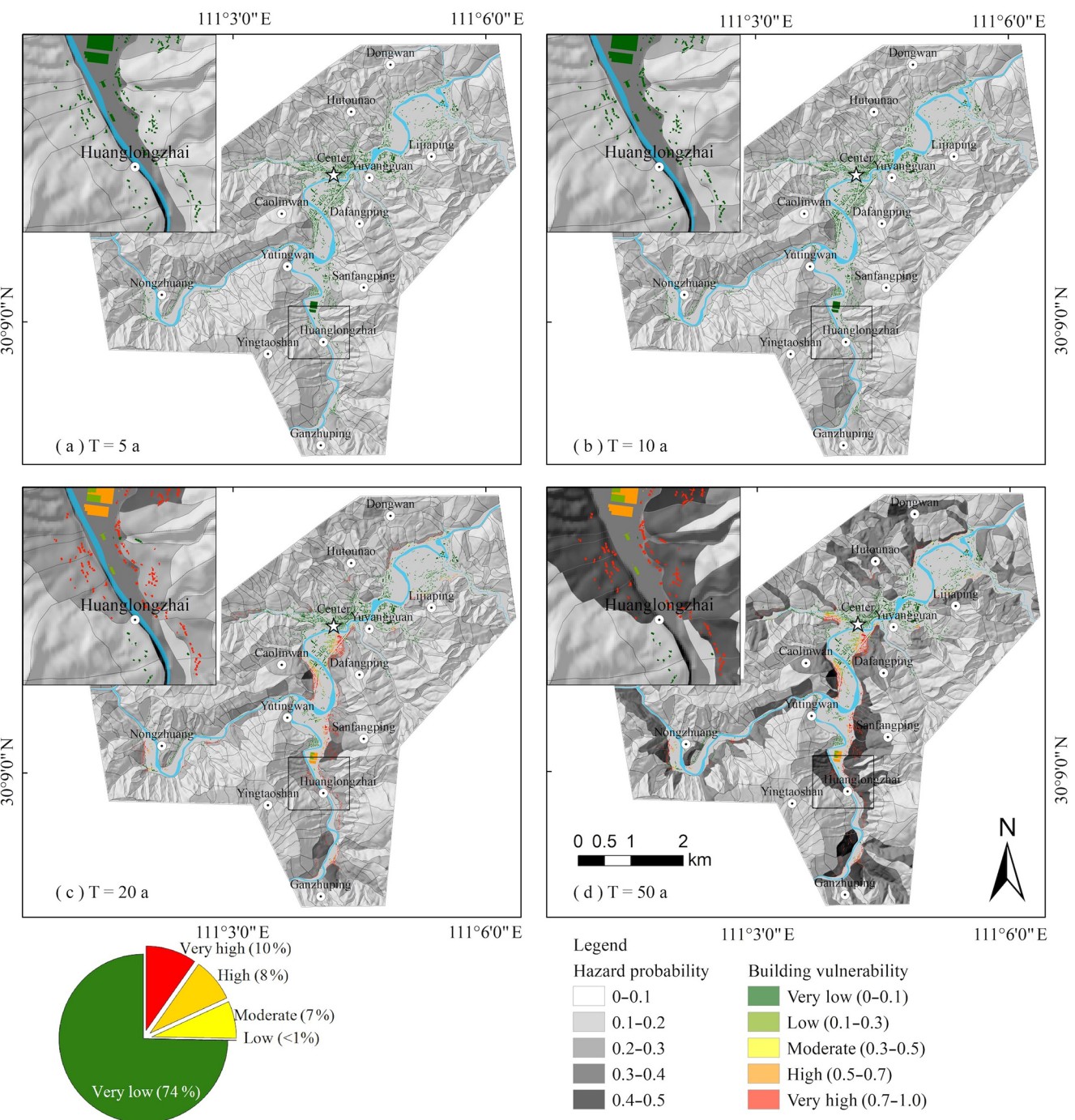

**Figure 13.** Buildings exposed to landslides, vulnerability distribution map for four return periods (5, 10, 20, and 50 years), and size scenario of landslide volume equal to or greater than $50\,000\,\text{m}^3$.

## 6   Conclusion

We conducted a semiquantitative risk assessment for landslides at the community level based on the definition of landslide risk provided by Varnes (1984) and the IAEG. In our case study, we focused on the potential damage of buildings and loss of life for the population in the buildings. We generated a susceptibility map, eight hazard maps, eight vulnerability maps, and eight risk maps for four return periods (5, 10, 20, and 50 years) and two size scenarios (equal to or greater than $50\,000\,\text{m}^3$ and equal to or greater than $100\,000\,\text{m}^3$). The landslide susceptibility result was tested, and it yielded a success rate of 0.84, highlighting the im-

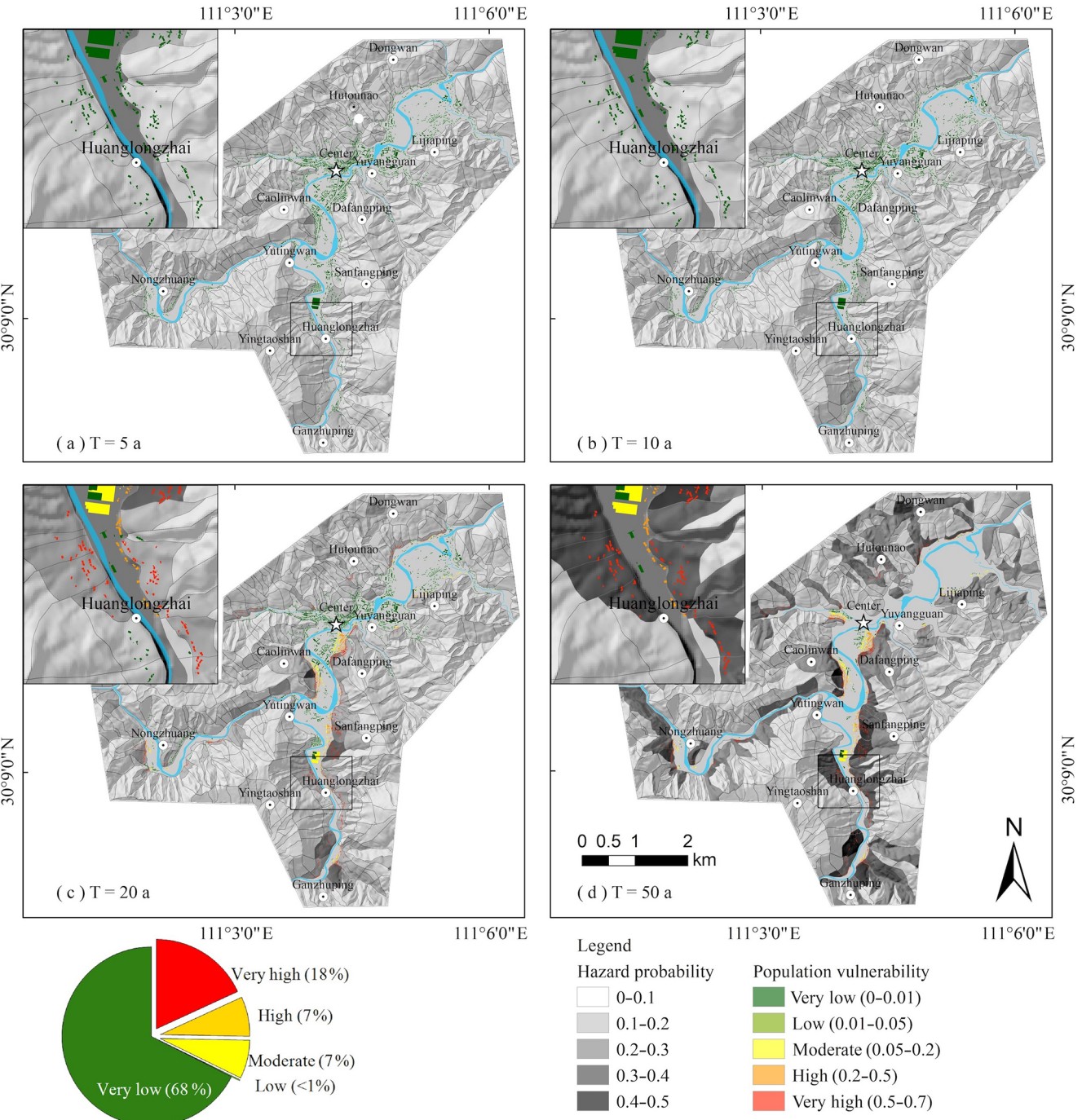

**Figure 14.** Population exposed to landslides and vulnerability distribution map for four return periods (5, 10, 20, and 50 years) and size scenarios of landslide volume equal to or greater than $50\,000\,\text{m}^3$.

portant contribution of $Q_4^{\text{d}_l+\text{e}_l}$ and the Guniutan $O_2g$ limestone. The approach for generating hazard maps, which involved integrating three probabilities (spatial, temporal, and size probabilities), proved applicable in the case study area. In the size probability calculation, the use of a normal distribution function for landslides requires caution, with a better fitting function suggested when small landslide data are scarce in an area. Also, the landslide influence area was empirically determined at a community level using simple data (slope height and volume) in the absence of geotechnical parameters. However, for a more accurate vulnerability assessment, numerical modeling on landslide travel distance is suggested because the resulting intensity parameters, such as velocity and depth, are essential input data for vulnerability

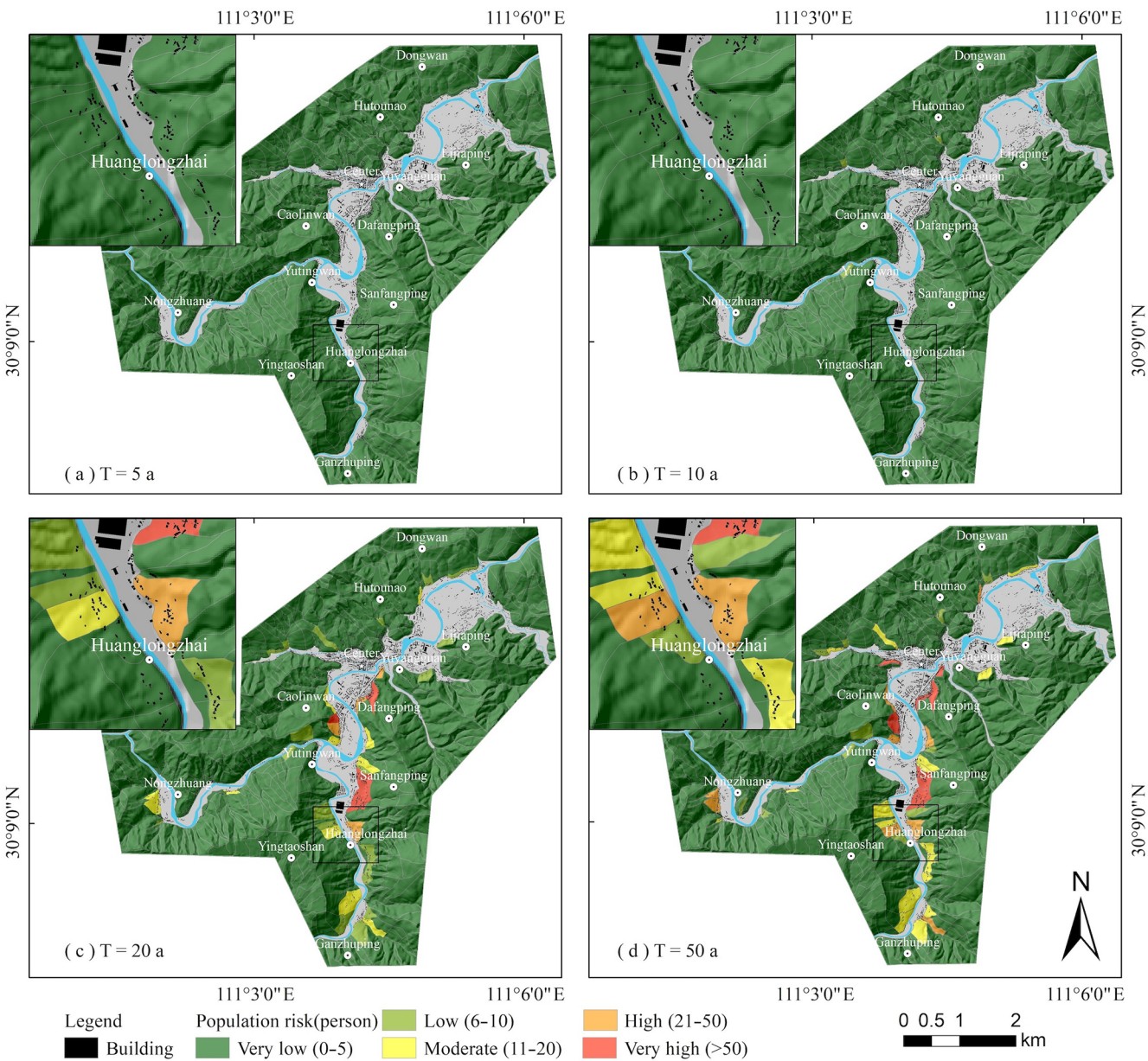

**Figure 15.** Example of a risk map for population loss for four return periods (5, 10, 20, and 50 years) and the landslide size scenario equal to or greater than $50\,000\,\mathrm{m}^3$.

quantification. Besides the presented limitations, we believe that the proposed risk maps can help local stakeholders in establishing periods of risk planning for the community, including short-term (in 5 years), mid-term (10–20 years), and long-term (50 years) strategies, or provide a reference for the cost–benefit analysis for each slope unit from the quantified annual risk values.

*Data availability.* The data are available upon request to the corresponding author.

*Author contributions.* KY, LC, YX, ZL, and DL supervised the field work and collected the complicated data. LC and SF discussed the plan for this article and designed and implemented all the experiments. SF compiled all data and prepared the draft including figures in the article. LC and TW revised the article while JD and CZ provided support on methods.

*Competing interests.* The authors declare that they have no conflict of interest.

*Special issue statement.* This article is part of the special issue "Advances in extreme value analysis and application to natural hazards". It is not associated with a conference.

*Acknowledgements.* We want to thank the editor and two anonymous reviewers for their constructive comments, which helped us improve the quality of the paper.

*Financial support.* This research has been supported by the National Natural Science Foundation of China (grant nos. 41877525, 41601563, and 41641012) and the Research Foundation of Guideline for Geological Hazards in Mountainous Communities in Wuling Area, China Geological Survey (grant no. D5.7.3).

*Review statement.* This paper was edited by Yasser Hamdi and reviewed by two anonymous referees.

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

**Remarks from the language copy-editor**

CE1  As we are only allowed to insert technical and necessary changes at this stage, the change you requested to the characteristics of lithology (replacing "Upper Carboniferous... strip limestone." with "Upper nodular limestone, lower limestone with cherts.") needs to be approved by the editor. Please provide an explanation for the editor for this change.

CE2  Please verify that the correct figure is being used.