# Peer review of "Community-based landslide hazard probability and risk assessment: A case in west Hubei, China"

_Natural Hazards and Earth System Sciences, 2019_

## Referee Comment (RC1) · Anonymous Referee #1 · 9 Oct 2019

A. General comments This manuscript presents a work regarding the landslide hazard and risk assessment of landslides in mountainous terrains of the Hubei Province (China). The study area is called Yuyangguan community which spreads over 34 km2. The authors applied a probabilistic method from different maps and datasets to propose landslides hazard, vulnerability and risk maps for four return periods (5, 10, 20, 50 years) and landslides size scenarios equal or greater than 50 000 m3. After the presentation of the introduction and context in the Yuyangguan area, the used methodology depicted in the part 3 is the heart of this project. The results are presented in part 4 and discussed in part 5 before the conclusion presented in the last part.

This manuscript represents a considerable work of analysis according to a methodology developed and supported by different bibliographical works. All the results are

rather convincing however the methodology was not reviewed in detail (part 3) and might require a further examination. I consider this contribution as can very useful and replicable in engineering practices to control the risks and to ascertain a sustainable development for this kind of geological context with frequent landslides. On the other hand, there are a few points that need clarification or complementary developments. In addition, some figures (maps) should be improved to appreciate the quality of this work.

B. Specific comments

1) It needs to present in more details the work on the field done by the authors to analyze the different causes of landslides (lithology, slopes, etc..). The authors present only 2 examples (figures 2 and 3) with field analysis.

In particular they do not talk enough about Chengguan area which represents the most historical cases in their database. Their approach seems to illustrate more the Chengguan area than Yuyanggguan area indeed. That should be more highlighted in this manuscript. In addition, the Chengguan area shall be more detailed in terms of similarities of context with the Yuyanggguan area (geology, geomorphology, climate, etc) to support the analysis with both.

Related to this topic of historical landslides: - Figure 1: the location of Chengguan community does not sufficiently precise in regard to the location of Yuyangguan community. What is Wufeng (Fig 1b) with respect to Yuyangguan ? It is not clear enough for the reader. - Figure I and Table II: the localization of historical landslides is not provided on the figure 1. In addition and the coordinates of each landslide in the table II should be added maybe.

2) This analysis on the field of historical cases is used to discuss and support the landslide susceptibility result in the paragraph 5.1 (Discussion on landslide susceptibility map). However the authors should develop also: a. The description with more details about the observed lithology on the field (like the most important controlling factor);

b. The structural control (fault, joints) which plays also a potential part in the cause of some landslides (aggravating factor). It is not sufficiently discussed if we note the presence of numerous faults in the study area indicated on the Figure 1 c.

Other point related to this topic, the authors have not mentioned the potential earthquake source (other triggering factors). If it's not relevant in this zone then it must be at least mentioned and discarded. Related to this point, the tectonic context should be added in the presentation of the geological context too brief in the manuscript.

3) Hypothesis from lines from 215 to 216: the assertion "assuming that the past is the future", landslides in the study area will probably occur with the same amount of landslides over the next 50 years as the past 50 years" is not sufficiently discussed and argued. In particularly the possibility of impacts of climate change (more heavy rains) should be included or at least introduced for the next 50 years like a limit or a next development to this study. This paragraph echoes to lines from 335 to 337 where the authors remind this assumption of a same condition between future and past to cause landslides. They indicate without details some possible changes of conditions but this issue deserves to be developed.

4) From line 105 to 109: Please develop, the explanation lacks of information. It needs to detail more ("Subsequently…..... in study area")

5) Line from 319 to 322: Would other factors exist to explain the difference with the classical distribution model (Malamud et al, 2004; Stark and Hovius, 2001) ?

6) The conclusion should be more developed about of limits and potential application of results.

C. Technical corrections

Figure 1: About faults on the figure 1c, could you indicate more information about the type of faults?

Figure 3: add scale into the zoom called "landslide surface"

Figures 4, 11, 12, 13 and 14: those maps are too small to be readable and impact the quality of this work. The names of villages or localities are difficult to read also.

Text:

From line 61 to 62: the main lithological units should be presented in the order of the geological ages.

From line 115 to 155: the methodology should be presented with more of clarity between each paragraph: determination of spatial probability (1), temporal probability (2) and size probability (3).

Line 199: rewrite and clarify the second part of this sentence "these two geological units can be susceptible to erosion and can quickly accelerate erosion ".

Line from 204 to 205: rewrite "the value of slope varies from 10°to 30° is 0.19"

Line from 304 to 305: rewrite, problem with the grammar sentence "This is because that although. . .., but the area. . ."

Line 314: The word compatible or suitable seems to be more adapted than "feasible"

Line 318: Bibliographical order according the growing age: 2001 before 2004. Review in the whole document.

---

## Short Comment (SC1) · 21 Oct 2019

Dear Referee, Thank you for your professional comments on our manuscript. These comments are all valuable and helpful for revising and improving our manuscript. Please note that the revised manuscript has been attached in the supplement file. The main corrections in the manuscript and the point-by-point responses to your comments are as following (the page number and line number in this letter refer to the revised manuscript):

Specific comments: (1) It needs to present in more details the work on the field done by the authors to analyze the different causes of landslides (lithology, slopes, etc. . .). The authors present only 2 examples (Figures 2 and 3) with field analysis. In particular

they do not talk enough about Chengguan area which represents the most historical cases in their database. Their approach seems to illustrate more the Chengguan area than Yuyangguan area indeed. That should be more highlighted in this manuscript. In addition, the Chengguan area shall be more detailed in terms of similarities to context with the Yuyangguan area (geology, geomorphology, climate, etc.) to support the analysis with both. Related to this topic of historical landslides: Figure 1: the location of Chengguan community does not sufficiently precise in regard to the location of Yuyangguan community. What is Wufeng (Fig.1b) with respect to Yuyangguan? It is not clear enough for the reader. Figure 1 and Table II: the localization of historical landslides is not provided on the figure 1. In addition and the coordinates of each landslide in the table II should be added maybe. Responses: Thank you very much for your comments. Due to the page limitation of the manuscript, we only show two landslides (Fig.2 and 3). These two are typical for the study area. Landslide in Fig. 2 represents slope instability by rainfall and Fig. 3 by slope cutting. Our case study area is Yuyangguan community, and the objective of the manuscript is to assess landslide risk for this community. Why we use the database of Chengguan is because the number of landslides in Yuyangguan is limited and not satisfying to calculate size probability. And also, it is because that the geology background and landslide type in Chengguan are similar to in the case community (Yuyangguan). So we didn't describe the details in Chengguan but just use the historical landslide data. We have complemented the statements from Line 94 to Line 98. Sorry for the confusion. In Figure 1, we pointed out the location of Chengguan. Sorry for the mistaken label (Wufeng) in Fig.1a, we have corrected it. We added the coordinates of each landslide in table II.

(2) This analysis on the field of historical cases is used to discuss and support the landslide susceptibility result in the paragraph 5.1 (Discussion on landslide susceptibility map). However the authors should develop also: a. the description with more details about the observed lithology on the field (like the most important controlling factor); b. the structural control (fault, joints) plays also a potential part in the cause of some landslides (aggravating factor). It is not sufficiently discussed if we note the

presence of numerous faults in the study area indicated on the Figure 1c. Other point related to this topic, the authors have not mentioned the potential earthquake source (other triggering factors). If it's not relevant in this zone then it must at least mentioned and discarded. Related to this point, the tectonic context should be added in the presentation of the geological context too brief in the manuscript. Response: It is a good comment, thank you! The observed lithology was described in line 61 to 64 and Table I. As to the geological structures, they influence the stability of slopes especially of rock slopes. As illustrated in Fig.1c, there are two landslides closing to the faults among seven landslides in red colors on the map. The majority of investigated landslides are soil or debris landslides in Table II, among which seven landslides are relatively small scales. It indicated that the fault has no significant impact to the landslides which have already occurred in this area. The manuscript has paid attention to the geological structure. Unfortunately, we lack sufficient discussions. So, we have added a necessary explanation to demonstrate the importance of these factors in the context. Please see line 65 to line 71. Actually, for regional-scale assessment, it is not easy to directly define the structural figures (faults, joints) as the factors such as defining the dip angle of faults or joints, etc. unless for a site-specific slope instability assessment. However, the manuscript has already designed an alternative index, distance to fault, to present the structures as in Fig.4 g comprehensively. For the tectonic factor, the studied area is located in a weak seismic activity region according to the assessment by the China Earthquake Administration. There were no historical records of earthquake-induced landslides in that area. More explanation has also been put in the context. Please see line 71 to line 74.

3) Hypothesis from lines from 215 to 216: the assertion "assuming that the past is the future", landslides in the study area will probably occur with the same amount of landslides over the next 50 years as the past 50 years" is not sufficiently discussed and argued. In particularly the possibility of impacts of climate change (more heavy rains) should be included or at least introduced for the next 50 years like a limit or a next development to this study. This paragraph echoes to lines from 335 to 337 where the

authors remind this assumption of a same condition between future and past to cause landslides. They indicate without details some possible changes of conditions but this issue deserves to be developed. Response: thank you very much for the suggestion. The assumption is very important for hazard probability calculation in this study. We have discussed this limitation in section 5.3. Further study and development are being taken in our research team now. But as you said, we did not sufficiently discuss the limitation. So we added more details about the possible changes in line 359 to line 361.

4) From line 105 to 109: Please develop the explanation lacks of information. It needs to detail more ("Subsequently......in study area") Response: thank you very much for the comment. We have added more detail information in this part from line 123 to line 127.

5) Line from 319 to 322: Would other factors exist to explain the difference with the classical distribution model (Malamud et al., 2004, Stark and Hovius, 2001)? Response: thank you very much for the suggestion. As illustrated in the context (line 338 to 341), no small landslides (< 1000 m2 in Malamud's research) in our study area is the main reason for the difference. Maybe landslide types and triggering factors are the other reasons for this difference. Further studies in comparison should be taken with more landslides events considering these factors. Complementary statements were added in line 342 to line 343.

6) The conclusion should be more developed about limits and potential application of results. Response: thank you very much for the comment. After carefully checking, we found that the limitations and potential application of results had been sufficiently pointed out in the conclusion part. Thank you for your reminding.

C. Technical corrections (1) Figure 1: About faults on the figure 1c, could you indicate more information about the type of faults? Response: thank you very much for the comment. We added the type and name of the fault in Fig.1c. And the more detailed

description was added in the context. Please see line 65 to line 72. (2) Figure 3: add scale into the zoom called "landslide surface". Response: thank you very much for the comment. We have added the scale bar into the figure called "landslide surface". (3) Figures 4,11,12,13 and 14: those maps are too small to be readable and impact the quality of this work. The names of villages or localities are difficult to read also. Response: thank you very much for the suggestion. We have enlarged the figures 4,11,12,13 and 14.

Text: (1) From line 61 to 62: the main lithological units should be presented in the order of the geological ages. Response: thank you very much for the suggestion. We have revised and presented the main lithological units in the order of the geological ages. Please see line 61 to line 62. (2) From line 115 to 155: the methodology should be presented with more of clarity between each paragraph: determination of spatial probability (1), temporal probability (2) and size probability (3). Response: thank you very much for the suggestion. We have clarified the methodology from line 115 to 155 by the third level section. Please see line 137 to line 165 (3) Line 199: rewrite and clarify the second part of this sentence "these two geological units can be susceptible to erosion and can quickly accelerate erosion". Response: thank you very much for the suggestion. We have rephrased this sentence. Please see line 221 to Line 222. (4) Line from 204 to 205: rewrite "the value of slope varies from 10°to 30° is 0.19". Response: thank you very much for the suggestion. We have rephrased this sentence. Please see line 226 to line 227. (5) Line from 304 to 305: rewrite, problem with the grammar sentence "This is because that although . . ., but the area. . .". Response: thank you very much for the suggestion. We have rephrased this sentence. Please see line 322. (6) Line 314: The world compatible or suitable seems to be more adapted than "feasible". Response: thank you very much for the suggestion. We have revised this sentence using the word "suitable". Please see line 335. (7) Line 318: Bibliographical order according the growing age: 2001 before 2004. Review in the whole document. Response: thank you very much for the comment. We have reviewed the whole document very carefully and revised them.

We tried our best to improve the manuscript and made some changes in the manuscript. We feel great thanks for your professional review work on our article, and hope that the correction and response will meet with approval.

Looking forward to hearing from you.

Sincerely, Lixia Chen

Please also note the supplement to this comment:
https://www.nat-hazards-earth-syst-sci-discuss.net/nhess-2019-259/nhess-2019-259-SC1-supplement.pdf
* * *
[Figure]

**Supplement:**

[revised manuscript text omitted]

The main lithological units outcropping in the study area comprise of Ordovician limestone with shale, Silurian sandstone
and shale, Devonian with coal layer, Permian sandstone and recent deposits. Silurian sandstone is the dominant rock
distributed along the community slopes. Intense weathering causes the bedrock to have low mass strength. Weathered rocks
in Silurian are the primary source of landslides.

65  Two main faults constrain the bulk architecture of the area, namely the Xiannvshan fault (Fault 1 in Fig. 1) and the
Yuyangguan fault (Fault II in Fig. 1). The Xiannvshan fault, striking approximately N340-345 °W and dipping 60-70 ° to the
northeast, is a trans-pressional fault terminating at the center of the community. This fault is a seismic activity belt,
represented by Panjiawan earthquake (Ms = 4.9) in 1961 and Zigui earthquake (Ms = 3.3) in 1972. The Yuyangguan fault,
striking east-west and dipping 60-70 ° to the south, is a trans-tensional fault. Fault rocks of the Yuyangguan fault are

composed of 20-50 m width of cataclasite and brecciated mylonite. Several secondary faults merged into the Yuyangguan fault and together go through the centre of the Yuyangguan community. According to the China Earthquake Administration, the studied area is located in a weak seismic activity region with basic earthquake intensity of Ⅵ. The 50-year 10% probability exceedance of peak ground acceleration is 0.05 g. There were no historical records of earthquake-induced landslides in the area.

[revised manuscript text omitted]

Table II. Historical landslides investigated in the field by the authors

| Landslide ID | Coordinates | | Material | Bedrock | Volume (×10 000 m³) | Area (×10 000 m³) | Date (Year.month) | Triggering factor |
|---|---|---|---|---|---|---|---|---|
| | X | Y | | | | | | |
| Seven landslides were reported in Yuyangguan community | | | | | | | | |
| YYG01 | 506692 | 3334830 | Soil | $S_2s$ | 147.6 | 5.9 | 2013.7 | R |
| YYG02 | 506615 | 3335500 | Soil | $S_1lr$ | 49.8 | 4 | 1981.6 | R & SC |
| YYG03 | 506828 | 3337320 | Soil | $S_1l$ | 20.9 | 1.7 | 2014.1 | R & SC |
| YYG04 | 506275 | 3338610 | Soil | $S_2s$ | 10.7 | 1.5 | 2013.5 | R & SC |
| YYG05 | 505788 | 3338880 | Soil | $S_2s$ | 10.4 | 3.5 | 1989.1 | R |
| YYG06 | 507320 | 3340080 | Soil&rock | $O_2g$ | 8 | 1.6 | 2007.12 | R & SC |
| YYG07 | 507381 | 3341680 | Rock | $O_2g$ | 22.2 | 2.2 | 2010.6 | R & SC |
| Nineteen landslides were reported in Chengguan community | | | | | | | | |
| CG01 | 466943 | 3339730 | Soil | $S_1l$ | 5.4 | 0.45 | 2009.5 | R |
| CG02 | 467038 | 3339980 | Soil | $S_1l$ | 1.05 | 0.21 | 2012.4 | R |
| CG03 | 467576 | 3340630 | Rock | $O_2b$ | 1.92 | 0.48 | 1991.7 | R |

| | | | | | | | | |
|---|---|---|---|---|---|---|---|---|
| CG04 | 467659 | 3340830 | Rock | $O_2b$ | 5.04 | 0.56 | 2005.8 | SC |
| CG05 | 467830 | 3340990 | Soil | $O_2b$ | 55.9 | 4.3 | 2002.7 | R & SC |
| CG06 | 468072 | 3341460 | Rock | $O_2b$ | 34.5 | 2.3 | 1992.6 | R |
| CG07 | 468295 | 3341510 | Soil | $O_2b$ | 14.3 | 1.1 | 1997.7 | R |
| CG08 | 468193 | 3341670 | Soil | $O_2b$ | 79.5 | 5.3 | 1967.6 | R |
| CG09 | 468331 | 3341980 | Soil&rock | $O_2b$ | 16.8 | 2.4 | 1969.7 | R |
| CG10 | 468597 | 3342480 | Soil | $O_2b$ | 3.18 | 0.53 | 2012.3 | R |
| CG11 | 468841 | 3342740 | Soil | $S_1l$ | 7 | 0.5 | 1992.6 | R |
| CG12 | 468354 | 3342720 | Soil | $S_1l$ | 9.6 | 1.2 | 1992.6 | R |
| CG13 | 467816 | 3342670 | Soil | $S_1l$ | 4.55 | 0.65 | 1994.7 | R |
| CG14 | 467888 | 3342520 | Soil | $S_1l$ | 8.2 | 0.82 | 1994.7 | R |
| CG15 | 467956 | 3342230 | Soil | $S_1l$ | 8 | 0.8 | 1998.2 | R |
| CG16 | 468188 | 3342070 | Soil | $S_1l$ | 3.6 | 0.6 | 1996.7 | R |
| CG17 | 468119 | 3342020 | Soil | $S_1l$ | 4 | 0.5 | 1969.7 | R |
| CG18 | 467707 | 3341990 | Soil | $S_1l$ | 13.2 | 1.1 | 2007.7 | R |
| CG19 | 467889 | 3341710 | Soil | $S_1l$ | 4.32 | 0.54 | 1991.6 | R |

R-rainfall; SC- Slope cut. The code of Bedrock is listed in Table I.
Projected Coordinate System is Xian_1980_3_Degree_GK_Zone_37 with the projection of Gauss_Kruger.

Note: Database of Chengguan community is complemented for hazard probability analysis because (1) the number of landslides in the case community is limited for probability analysis; and (2) these two communities have 
[revised manuscript text omitted]

---

## Referee Comment (RC2) · Anonymous Referee #2 · 14 Nov 2019

General comments

First, it is important to clarify that the two versions of the manuscript were reviewed (August 23rd and October 22nd) and I can affirm that the manuscript improved largely in the second version. My comments refer to the version contained in the document named "nhess-2019-259-AC1-supplement".

The manuscript contains the results of a solid and detailed research on hazard and risk at a local scale in the Hubei province in China.

In general terms, the ideas of the authors are clear. However, the document presents several grammatically awkward sentences, expressions that should be checked and lacks connections among the sentences. Therefore, proofreading and English editing

services are recommended to ensure that the English of the manuscript is up to the publication standard.

Specific comments The term Community-based leads the reader to think that the performed hazard and risk analysis involved members of the local community as part of a participatory methodology. In this case, it seems that you use the term "Community-based" to refer to the scale of the analysis, more than the participants of the process. I would suggest to change the title accordingly. For example, in the introduction you used the more appropriate term "community-level", or you could use something like local -level, community scale, or any other term that refers to the scale of your work.

In general, when using the term community, it is not always clear if it refers specifically to the Yuyangguan community, or if it is used to refer to a local scale. This creates confusion among the reader so I would suggest to check the manuscript and clarify this when necessary.

The introduction contains all the necessary information. However, the current order generates confusion among the readers. I would suggest to switch the first and second paragraph and include some connectors. Additionally, the use of the future sentence should be avoided.

Considering that readers might not be familiar with China's administrative division and geography, the study area description should be complemented with basic geographic and administrative information. It is clear that the area is in the Hubei province, but is not clear in which Prefecture and/or County the communities are located,and what villages the selected communities include. Additionaly, it would be interesting to mentioned when the area started to be inhabited and if there has been any recent intense urbanization development, considering that one of the landslide triggers are man-made actions, such as roads construction.

Since in some points you focused only in Yayangguan community and in some others, you complement it with information of and Chengguan community (i.e. line 175), I

would suggest to clarify since the beginning what is the goal and scale of the analysis in both communities.

In the methodology section I suggest to include a figure with a conceptual map of the different components of the hazard and risk analysis process.

Technical corrections

The last sentence of the introduction is confusing "This achievement may also be utilised into community scale landslide risk assessment in a mountainous area in Hubei, China." I would suggest you to check the whole paragraph in order to articulate the different sentences, while avoiding repetition.

In Fig. 1, please clarify what does each inset represent at administrative or geographical level. Inset (b) seems like a province but it looks deformed, and is not clear what the limits of inset (c) correspond to. Maybe you can try by adding thin administrative division lines at prefecture level. Additionally, it is recommended to include a frame in the bigger map to show where the zoom area is located (as done in Fig. 8).

Line 63, check the phrase "Weathered rocks in Silurian are the primary source of landslides.", maybe you mean something like "As a consequence, weathered Silurian rocks are the primary source of landslides."

Line 64, the sentence "Weathered rocks in Silurian are the primary source of landslides" seems to contradict the affirmation of the introduction regarding that " that Quaternary deposits and weathered eluvium from Ordovician limestone are the two major controlling factors." This confusion is presented again in the first paragraph of the section 5.1.

Line 67, check the phrase "This fault is a seismic activity belt,", seems to refer more to "This fault is a seismically active belt"

Line 75, check the phrase "Rainfall and human activities contribute significantly to the slope movement", could be something like "Rainfall and human activities contribute

significantly to trigger mass movements. . ."

Line 75, clarify that the mentioned landslides are just examples and connect this to the following paragraph.

All maps. Yayangguan and Chengguan communities should be labelled in all the maps

Figure 5. Please remove the labels that do not correspond to locations, but to specific buildings, ex. School, health center, etc.

Fig.5, 8, 11 it is not clear why the zoom frame shows the area around Huanglongzhai, instead that Yayangguan or Chengguan

Table II. I would suggest to organize the landslides by date, in order to allow the reader to have an idea of recurrence.

Line 148, explain what does W and W+ correspond to.

Line 238. I suggest to support the affirmation"Assuming that the past is the future, landslides in the study area will probably occur with the same amount of landslides over the next 50 years as the past 50 years." That, considering Climate change, but also the anthropic incidence as landslide trigger. That affirmation depends largely in the history of the area, i.e. if the area has sustained a stable urban development in the last 50 years the affirmation is valid, but if it is not the case, then the affirmation should be supported with strong arguments.

Line 253. Please include some arguments to support the decision of using 50 000 cubic meters and 100 000 cubic meters for the size scenarios, since, according to Table II, there have been historic landslides with a considerably larger volume.

Fig. 10 (b). Correct the label of Number of landslides

Line 322, please check the wording of the last sentence of this paragraph.

Line 360, maybe you refer to "the number of landslides", instead than "the number of

historical". Please check

Line 364, regarding the affirmation "...risk management on slope units with very high-class hazard probability and very high-risk can be suggested as relocation or engineering works," the decision to relocate people is a delicate one, and should be taken based in a large array of information, not solely in risk maps of not detailed scale. In this case, I would suggest to review the affirmation and to propose a more detail geotechnical analysis of the very high-class areas, instead than inviting to relocate the people based solely in these results.

Line 368. Check the sentence coherence.

Best regards

---

## Author Comment (AC2) · 18 Nov 2019

Dear Referee,

Thank you very much for your professional comments on our manuscript. These comments are all valuable and helpful for revising and improving our manuscript.

To keep manuscript turnover times low, we respond to you how we plan to revise and what we have done. We will submit the revised manuscript later as soon as possible.

General comments

First, it is important to clarify that the two versions of the manuscript were reviewed (August 23rd and October 22nd) and I can affirm that the manuscript improved largely

in the second version. My comments refer to the version contained in the document named "nhess-2019-259-AC1-supplement".

The manuscript contains the results of a solid and detailed research on hazard and risk at a local scale in the Hubei province in China.

In general terms, the ideas of the authors are clear. However, the document presents several grammatically awkward sentences, expressions that should be checked and lacks connections among the sentences. Therefore, proofreading and English editing services are recommended to ensure that the English of the manuscript is up to the publication standard.

Responses: Thank you very much for your comments. We will make great efforts to improve the language as best as we can by proofreading and English editing services, sorry for the grammatically awkward sentences and expressions.

Specific comments:

(1) The term Community-based leads the reader to think that the performed hazard and risk analysis involved members of the local community as part of a participatory methodology. In this case, it seems that you use the term "Community based" to re-fer to the scale of the analysis, more than the participants of the process. I would suggest to change the title accordingly. For example, in the introduction you used the more appropriate term "community-level", or you could use something like local -level, community scale, or any other term that refers to the scale of your work.

In general, when using the term community, it is not always clear if it refers specifically to the Yuyangguan community, or if it is used to refer to a local scale. This creates confusion among the reader so I would suggest to check the manuscript and clarify this when necessary.

Responses: Thank you very much for your comments and suggestion. We used the term "community-based" in the manuscript to refer to the scale. In Chinese language

the term "community-based" can refer to two senses: scale and administration. Sorry for the confusing words from Chinese to English. Community level is a good suggestion. We will change the title as 'Landslide hazard probability and risk assessment at the community level: A case in west Hubei, China'.

(2) The introduction contains all the necessary information. However, the current order generates confusion among the readers. I would suggest to switch the first and second paragraph and include some connectors. Additionally, the use of the future sentence should be avoided.

Responses: Thank you very much for your comments and suggestion. We will re-manage the order of the paragraphs. The future sentence and related language mistakes will be carefully checked and avoided.

(3) Considering that readers might not be familiar with China's administrative division and geography, the study area description should be complemented with basic geographic and administrative information. It is clear that the area is in the Hubei province, but is not clear in which Prefecture and/or County the communities are located, and what villages the selected communities include. Additionally, it would be interesting to mention when the area started to be inhabited and if there has been any recent intense urbanization development, considering that one of the landslide triggers are man-made actions, such as roads construction.

Responses: Thank you very much for your comments and suggestions. We complement some necessary information about the geography and villages in the selected community. Meanwhile, it is a very good suggestion to state the urbanization of the community, which is very helpful for readers to understand the triggers of the landslides. We will revise the first paragraph in part 2 as: The current study area, Yuyangguan community (N 30°03′- 330°15′, E 110°08′- 8110°08′), is located in Wufeng county, western Hubei province, China (Fig. 1). It covers an area of about 34 km2 and includes 15 villages, such as Sanfangping, Dafangping, Caojiaping etc. The

community started to be inhabited about 750 years ago, and intense urbanization development and the national road construction crossing the community began in 2012. The study area was selected due to frequent landslide activities and caused subsequent damages in recent years. The residential area is surrounded by steep slopes, with an elevation of 180 to 680 m.a.s.l. The climate is characterized as a typical monsoonal climate, with an average annual precipitation of about 1500 mm.

(4) Since in some points you focused only in Yuyangguan community and in some others, you complement it with information of and Chengguan community (i.e. line 175), I would suggest to clarify since the beginning what is the goal and scale of the analysis in both communities.

Responses: Thank you very much for your suggestions. The objective of the manuscript is to assess landslide risk for Yuyangguan community. But when we analyze the size probability, it is found that the samples of the historical landslides are not sufficient for statistical analysis in size probability. So, we complement some other samples from the neighboring Chengguan community, considering that historical landslides and geological environment are similar in both two communities. We will complete more detailed information about the similarities in Line 98 of the manuscript version of 'nhess-2019-259-AC1-supplement': Administratively, these two communities are in one county called Wufeng. Geologically, both communities belong to the same geological structure belt named Changleping anticline fold which extends nearly east-western direction. The formation consists of the Silurian and Ordovician rocks. Landslides occurred in these two communities are same types as shallow and soil slope movement in weathered bedrock. The triggering factors of the landslides are rainfall and slope cutting in urbanization development.

(5) In the methodology section I suggest to include a figure with a conceptual map of the different components of the hazard and risk analysis process.

Responses: Thank you very much for your suggestions. We will include a figure for

the methodology as following attachment of Fig. 7 Flow chart of the methodology for landslide hazard risk assessment.

Technical corrections

(1) The last sentence of the introduction is confusing "This achievement may also be utilized into community scale landslide risk assessment in a mountainous area in Hubei, China." I would suggest you to check the whole paragraph in order to articulate the different sentences, while avoiding repetition.

Responses: Thank you very much for your comments. We checked the whole paragraph and agree with you to avoid repetition. The whole manuscript will be improved in language and expressions.

(2) In Fig. 1, please clarify what each inset represents at administrative or geographical level. Inset (b) seems like a province but it looks deformed, and is not clear what the limits of inset (c) correspond to. Maybe you can try by adding thin administrative division lines at prefecture level. Additionally, it is recommended to include a frame in the bigger map to show where the zoom area is located (as done in Fig. 8).

Responses: Thank you very much for your comments and suggestions. Insert (b) is a county called Wufeng, which includes the studied community Yuyangguan. Insert (c) is a zoomed area (the study area of Yuyangguan community) from insert (b). Sorry for the confusing. We added some necessary lines in Fig 1 (please see the attachment of Fig 1).

(3) Line 63, check the phrase "Weathered rocks in Silurian are the primary source of landslides", maybe you mean something like "As a consequence, weathered Silurian rocks are the primary source of landslides."

Responses: Thank you very much for your comments and suggestions. The meaning here is what you understand. We will revise this in the manuscript as: As a consequence, weathered rocks in Silurian and Ordovician (O2g) are the primary source of

landslides, the records of which and their bedrocks are shown in Table II.

(4) Line 64, the sentence "Weathered rocks in Silurian are the primary source of land-slides" seems to contradict the affirmation of the introduction regarding that "that Quaternary deposits and weathered eluvium from Ordovician limestone are the two major controlling factors." This confusion is presented again in the first paragraph of the section 5.1.

Responses: Thank you very much for your comment. We observed in the field that the source of majority landslides is from the weathered rocks in Silurian, and only two landslides in Yuyangguan are from weathered limestone in Ordovician. But in susceptibility analysis, we find that the weight of weathered rocks in Silurian is not the highest. It is because that the frequency of landslides in Silurian is lower than it in Ordovician, due to the area of Silurian in the map is much larger than the area of Ordovician. We will clarify the description with more details. Sorry for the confusing.

(5) Line 67, check the phrase "This fault is a seismic activity belt", seems to refer more to "This fault is a seismically active belt"

Responses: Thank you very much for your comment. The phrase in the manuscript will be changed as 'This fault is a seismically active belt'.

(6) Line 75, check the phrase "Rainfall and human activities contribute significantly to the slope movement", could be something like "Rainfall and human activities contribute significantly to trigger mass movements…"

Responses: Thank you very much for your comment. The phrase in the manuscript will be changed as 'Rainfall and human activities contribute significantly to trigger mass movements…'.

(7) Line 75, clarify that the mentioned landslides are just examples and connect this to the following paragraph.

Responses: Thank you very much for your comment and suggestion. One more sentence will be added in the end of this paragraph: These two landslides are examples, and more landslide records are prepared by aerial photos interpretation and then validated in field investigation.

(8) All maps. Yuyangguan and Chengguan communities should be labelled in all the maps.

Figure 5. Please remove the labels that do not correspond to locations, but to specific buildings, ex. School, health center, etc.

Fig.5, 8, 11 it is not clear why the zoom frame shows the area around Huanglongzhai, instead that Yuyangguan or Chengguan

Responses: Thank you very much for your comment. Because of the non-sufficient description on the studied community, the following figures in the manuscript gave you a confused impression. Figures of 5, 8 and 11 are for the area of Yuyangguan community, the zoom frame in which shows an area with high risk slopes and buildings. Sorry for the confusing and we will revise all the figures in labels after carefully checking.

(9) Table II. I would suggest to organize the landslides by date, in order to allow the reader to have an idea of recurrence.

Responses: Thank you very much for the good suggestion. We will re-organize the landslides by date.

(10) Line 148, explain what W- and W+ corresponds to.

Responses: Thank you very much for the good comment. We will add the explanation for W- and W+ as: In WoE method, W+ and W- were used as the weights where the evidence was present or absent respectively, thus the contrast (difference of W+ and W-) was used as a weight for each morphometric or geo-environmental factor.

(11) Line 238. I suggest to support the affirmation "Assuming that the past is the future, landslides in the study area will probably occur with the same amount of landslides over

the next 50 years as the past 50 years." That, considering Climate change, but also the anthropic incidence as landslide trigger. That affirmation depends largely in the history of the area, i.e. if the area has sustained a stable urban development in the last 50 years the affirmation is valid, but if it is not the case, then the affirmation should be supported with strong arguments.

Responses: Thanks for your very good comments. We are agreed with your idea that the affirmation depends largely in the history of the area. While dealing with the landslide issues, the hazards are increasingly connected with human activities and climate change which are becoming as the active triggering factors. If we purely consider the landslide from controlling geological factors, the sense of "assuming the past is the future" seems to be reliable during short period of decades or centuries comparing to geological time scale. In order to simplify the hazard probability calculation in this study, we used this assumption even it has limitation for the future. Anyway, the expression of the assumption is too deterministic, so we modified the expression softer as "assuming the past reflects the future". We have also discussed this limitation in section 5.3. More details are added about the possible changes in line 359 to line 361. Of course, further study and development are being taken in our research now following your constructive comments.

(12) Line 253. Please include some arguments to support the decision of using 50 000 cubic meters and 100 000 cubic meters for the size scenarios, since, according to Table II, there have been historic landslides with a considerably larger volume.

Responses: Thanks for your very good comments. We will implement some sentences at the end of the paragraph: In fig. 10b, we find that the number of landslides begins to increase rapidly when the volume is greater than 50 000 cubic meters, which represents a probable threshold value for landslide hazard volume. While, the value of landside volume larger than 100 000 cubic meters can represent the majority number of landslides as shown in Fig. 10b. Also, the value of 100 000 cubic meters is a standard threshold for landslide classification provided by China Geology Survey and

widely accepted value in landslide hazard risk control in China.

(13) Fig. 10 (b). Correct the label of Number of landslides

Responses: Thanks for your comment. We will correct the label of 'Number of land-slides'.

(14) Line 322, please check the wording of the last sentence of this paragraph.

Responses: Thank you very much for the comment. We will carefully revise the last sentence of this paragraph in grammar and wording.

(15) Line 360, maybe you refer to "the number of landslides", instead than "the number of historical". Please check.

Responses: Thank you very much for the comment and suggestion. We will add the missing word "landslide".

(16) Line 364, regarding the affirmation ". . . risk management on slope units with very high-class hazard probability and very high-risk can be suggested as relocation or engineering works," the decision to relocate people is a delicate one, and should be taken based in a large array of information, not solely in risk maps of not detailed scale. In this case, I would suggest to review the affirmation and to propose a more detail geotechnical analysis of the very high-class areas, instead than inviting to relocate the people based solely in these results.

Responses: Thank you very much for your comments. We totally agree with you that the decision of relocation or engineering works on slopes with very high-class hazard probability and very high-risk must be considered based on large array of information. The regional risk zonation is informative for planning. But for specific site treatment, it is essential to complement geotechnical analysis which gives strong support for decision-making. So, we have corrected expression in the context.

We will revise the last paragraph in section 5.3 as: From susceptibility to risk assessment, in general it is assumed that landslide will occur under the same condition where historical landslides occurred. However, morphometric, geo-environmental conditions will change with time. For example, the slope degree, elements at risk and land use cover may change because of the new infrastructures, or the number of historical landslides will increase due to heavier rainfall. Future studies are necessary in terms of the changing conditions for more accurate results and more practical applications. So far, the series of maps can be referred as the basis for landslide risk control and land use planning in Yuyangguan community. Risk controlling measurements can be planned on each slope unit using a matrix from the combination of landslide hazard probability and risk maps. For example, risk management on slope units with very high-class hazard probability and very high-risk can be suggested as relocation or engineering works. The precondition is that more detailed geotechnical investigation and comprehensive analysis is complemented. While, the slopes with high-class hazard probability and mid-class risk are suggested as monitoring programs if it is important for the safety of the surrounding population, roads, or other elements at risk. Meanwhile, the results of the four return periods and two size scenarios can be used for multi-temporal land use planning, such as short term (in 5-year), mid-term (10-20-year), and long-term (50-year). The annual risk value on each slope can be used for cost-benefit analysis in risk decisions. Meanwhile, the studies answer the question raised by Guzzetti et al. (2005) that how to combine a large number of hazard maps efficiently for different users.

(17) Line 368. Check the sentence coherence.

Responses: Thank you very much for your comments. We will carefully check the sentence coherence and grammar.

We tried our best to improve the manuscript and will make changes in the manuscript. We feel great thanks for your professional review work on our article, and hope that the corrections and responses will meet with approval.

Looking forward to your reply.

Sincerely, Lixia Chen

**Geological environment**

- Terrain
- Lithology
- Geology structure

**Landslide inventory**

- Date of occurrence
- Size
- Triggers

**Other data**

- Building footprint map
- Human activity / census data

Weight of evidence model → Grid-based landslide susceptibility mapping

Poisson model

Partition method of slope unit

Ordinary least square method

Slope unit based landslide susceptibility mapping

Landslide hazard mapping ← Spatial probability × Temporal probability × Size probability

Building material

Power-law relationship between population & building

Travel distance

Dasymetric method

Hazard × Vulnerability × Elements at risk → Landslide risk mapping

**Fig. 1.** Fig. 7 Flow chart of the methodology for landslide hazard risk assessment

(a)

Beijing

N

0 15 30 km

Chengguan Yuyangguan

(b)

**Legend**

Lithology

$Q^{al+pl}$

$Q^{dl+el}$

$P_1mn$

$P_1q$

$C_2hn$

$D_3x$

$D_3h$

$D_2y$

$S_2s$

$S_1lr$

$S_1l$

$O_2b$

$O_2g$

$O_1n$

$O_1h$

$O_1d$

$\in_2sn$

Fault

Landslide

Slope<5°

River

(c)

Two main faults:
Fault I - Xiannvshan fault
Fault II - Yuyangguan fault

111°3'0"E    111°6'0"E

30°12'0"N    30°12'0"N

Dongwan
YYG07

Fault I
65

Hutounao

Fault II

YYG06        Lijiaping

70          Center   Yuyangguan      70

Fault II

70

Caolinwan
YYG05        Dafangping
YYG04

Yutingwan

Nongzhuang                    Sanfangping
YYG03

Huanglongzhai
Yingtaoshan

YYG02

YYG01
Ganzhuping

30°9'0"N     30°9'0"N

0  0.5  1      2
km

N

111°3'0"E    111°6'0"E

**Fig. 2.** Fig. 1. a) Inset showing the map of China with the color (b) Representing the county which includes the case community Yuyangguan and the neighbor community Chengguan. c) Shows the distribution of lit

---

## Author Comment (AC3) · 2 Dec 2019

Dear Editor,

Here, we would like to submit our revised manuscript to for.
* * *